



# Rectified tidal transport in Lofoten-Vesterålen, Northern Norway

Eli Børve[1,2], Pål Erik Isachsen[1,3], and Ole Anders Nøst[4,5]

[1]Department of Geosciences, University of Oslo, 0315 Oslo, Norway
[2]Akvaplan-niva AS, 9296 Tromsø, Norway
[3]The Norwegian Meteorological Institute, 0371 Oslo, Norway
[4]Faculty of Biosciences and Aquaculture, Nord University, 8026 Bodø, Norway
[5]Akvaplan-niva AS, 7462 Trondheim, Norway

**Correspondence:** Eli Børve (elb@akvaplan.niva.no)

**Abstract.** Vestfjorden in Northern Norway, a major spawning ground for the Northeast Arctic cod, is sheltered from the continental shelf and open ocean by the Lofoten-Vesterålen archipelago. The archipelago, however, is well known for hosting strong and vigorous tidal currents in its many straits, currents that can produce significant time-mean tracer transport from Vestfjorden to the shelf outside. We use a purely tidally-driven unstructured-grid ocean model to look into nonlinear tidal

dynamics and the associated tracer transport through the archipelago. Of particular interest are two processes: tidal pumping through the straits and tidal rectification around islands. The most prominent tracer transport is caused by tidal pumping through the short and strongly nonlinear straits Nordlandsflaget and Moskstraumen near the southern tip of the archipelago. Here tracers from Vestfjorden are transported tens of kilometers westward out on the outer shelf. Further north, weaker yet notable tidal pumping also takes place through the longer straits Nappstraumen and Gimsøystraumen. The other main transport

route out of Vestfjorden is south of the island of Røst. Here the transport is primarily due to tracer advection by rectified anticyclonic currents around the island. There is also an anticyclonic circulation cell around the islands of Mosken-Værøy, and both cells have have flow speeds up to 0.2 m/s, magnitudes similar to the observed background currents in the region. These high-resolution simulations thus emphasize the importance of nonlinear tidal dynamics for transport of cod eggs and larvae in the region.

## 1   Introduction

Increased industrial activity along the Norwegian coast rises concern about potential impacts on the marine ecosystem. To properly assess risks involved, we need to understand oceanic dynamics in near-shore regions and its associated transport of nutrients and pollutants. Together with wind and freshwater run-off, tides often dominate the flow dynamics in coastal regions. While strong tidal currents are known to cause efficient vertical mixing of the ocean, important for bringing up nutrients to

the surface and ventilating the coastal seas, their contribution to net horizontal transport is often underestimated due to their oscillating nature. However, when strong tidal currents interact with complex topography in shallow waters, nonlinear flow dynamics can produce significant time-mean lateral transport (Huthnance, 1973; Parker, 1991).





In this study we will investigate nonlinear tidal dynamics around Lofoten-Vesterålen in Northern Norway (Figure 1), a major spawning ground for the Northeast Arctic cod (Hjermann et al., 2007). Spawning of this species takes place all along the middle and northern Norwegian coast, but as much as 40 percent of the cod spawns in Vestfjorden southeast of the Lofoten-Vesterålen archipelago (Ellertsen et al., 1981; Sundby and Bratland, 1987). The cod eggs and larvae spend the first five months drifting with the ocean currents from Vestfjorden to nursing grounds in the Barents Sea (Ådlandsvik and Sundby, 1994), and the survival rate during this initial pelagic drift is crucial for the recruitment of the fish stock (Hjort, 1914; Houde, 2008). Therefore a good understanding of drift and spreading patterns of the cod eggs and larvae is important for identifying particularly vulnerable regions and factors controlling the recruitment of the Northeast Arctic cod.

The majority of studies on transport of Northeast Arctic cod eggs and larvae have focused on flow dynamics on the Norwegian shelf where the Norwegian Coastal Current (NCC) and Norwegian Atlantic current (NwAC) quickly transport the cod larvae northeastward and into the Barents Sea (e.g. Ådlandsvik and Sundby, 1994; Vikebø et al., 2007; Opdal et al., 2008). The transport out of Vestfjorden itself has been reported to mainly take place around the southern tip of the Lofoten-Vesterålen archipelago (Vikebø et al., 2007; Opdal et al., 2008), following the larger-scale background currents, notably the NCC, and currents that respond to sporadic wind events. But by including stokes drift by wind-generated surface gravity waves, Röhrs (2014) and Röhrs et al. (2014) found that particles were transported closer to the coast and that the many straits which cut through the archipelago might be of larger importance than previously assumed.

The straits are well known for hosting strong and vigorous tidal currents. This includes a set of narrow and relatively long straits along the northern half of the archipelago, but even more so 2–3 wider but short straits over the shallow ridge southwest of Lofotodden (Moe et al., 2002). Here, near the southern tip of Lofoten, Moskstraumen is situated, also called the Lofoten maelstrom and famous for its vigorous and deadly currents. For the interested reader, tales, stories and observations of the Lofoten maelstrom can be traced all the way back to the medieval ages (see Gjevik et al., 1997). It seems clear that the vigorous tidal transport and dispersion around Moskstraumen in particular, but also in other straits of Lofoten, can impact the net exchanges between Vestfjorden and the shelf outside. Existing studies have focused on quantifying tidal dispersion rates (Lynge et al., 2010) and on establishing a link between tidal dispersion and transport by time-mean currents (Ommundsen, 2002). There has, however, been less attention put on identifying and quantifying the underlying non-linear dynamics responsible for tidal dispersion and transport. Two such nonlinear processes that are likely to be important in our region, and will therefore be the focus of the present study, are tidal pumping and tidal rectification.

Tidal pumping in a strait is a Reynolds flux of properties caused by a temporal asymmetry in circulation patterns between the flood and ebb phases of the tide (Geyer et al., 2001). The process can be explained using the simple model by Stommel and Farmer (1952), who were the first to investigate this phenomenon. When tidal currents enter a strait, say from the open ocean side, we expect them to behave roughly as potential flow and be steered by the coastline into the opening. So waters from a wide region around the opening, the 'sink region', is pulled into the strait. In contrast, when the flow exits the strait during the subsequent phase of the tide, the joint effect of friction and an adverse nonlinear pressure gradient as the strait opens up might cause the flow to separate from the coastline (Kundu et al., 2016). If there is such flow separation, the exiting water will continue straight ahead as a tidal jet. The areas covered by the sink region and the tidal jet are equally large, but they clearly



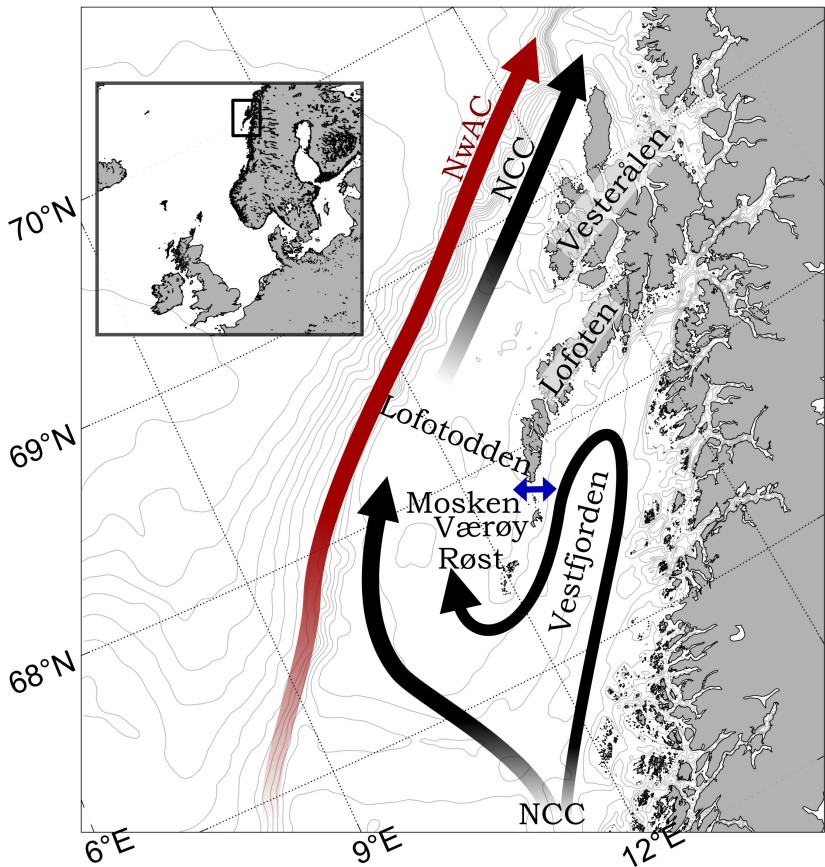

**Figure 1.** The general ocean surface circulation in the Lofoten-Vesterålen region. Black arrows show the Norwegian Coastal Current (NCC) and the red arrow shows the Norwegian Atlantic Current (NwAC). The blue two-headed arrow show the location of Moskstraumen, situated between Lofotodden to its north and the small island Mosken to its south.

take on different shapes. Some regions are overlapping while others are not. The existence of non-overlapping regions will cause some difference in what waters flow into and out of the strait. More recent studies have found that the presence of a tidal
jet on outflow from a strait is intimately related to the formation of self-propagating dipoles at the strait exit (Wells and van Heijst, 2003; Afanasyev, 2006; Nøst and Børve, 2021). The dipoles emerge from vortices that form at the points where the flow separates from the coastline, one at each side of the strait exit. The vortices become a self-propagating dipole when the strait is narrow enough for the two to interact, so that the velocity field of one vortex begins to advect the vorticity of the other. This self-propagating dipole is then trailed by the tidal jet. As it turns out, most of the water that exits the strait is injected into
the dipole and its trailing jet (Nøst and Børve, 2021). Therefore, if the dipole avoids being drawn back into the strait during the subsequent potential flow phase of the tide, the result will be a net property exchange through the strait (Kashiwai, 1984; Wells and van Heijst, 2003; Nøst and Børve, 2021).





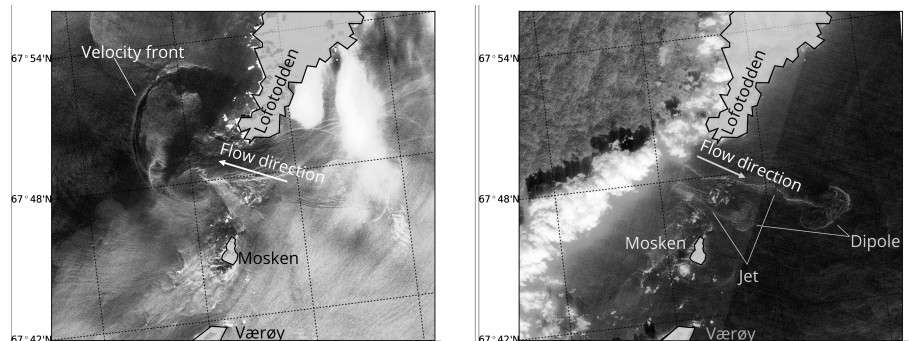

**Figure 2.** Satellite images from Copernicus Sentinel-II missions showing the surface currents in Mokstraumen and Nordlandsflaget.The Sentinel-II missions satellites carry a multi-spectral instruments with 13 spectral channels in the short wave infrared and visible/near infrared spectral range, whereas this image is collected from band B4 (664.6 nm). The satellite imagery was assessed and processes using data from the Norwegian National Ground Segment for Sentinel data (Halsne et al., 2019, pers. comm. Trygve Halsne).

The second process, rectification of oscillating currents around isolated islands and banks, has been observed in several regions where cross-slope tidal currents are prominent. The phenomenon can be explained as a response to a nonlinear mo-
mentum transport convergence by the oscillating currents (Huthnance, 1973; Loder, 1980) or, alternatively, a net cross-slope vorticity flux by the same oscillations (Zimmerman, 1978; Robinson, 1981). The generation of a net vorticity flux can be understood by imagining following a water column that moves periodically up and down the topographic slope of a bank, driven by a large-scale tidal potential (Zimmerman, 1978, 1981). In the northern hemisphere, the column attains negative vorticity on its way up the slope and positive vorticity on its way down due to vortex squeezing and stretching, respectively. Bottom friction
then removes some negative vorticity from the column over shallow regions and some positive vorticity over deep regions. A sustained oscillation, driven by the large-scale tidal potential, will hence be associated with a positive vorticity flux from shallow to deep regions. In a quasi-steady state, the vorticity flux from many such columns may be balanced by bottom friction acting on a time-mean anti-cyclonic circulation around the bank. Additionally, a net vorticity flux across a sloping bottom can be generated by differential bottom friction acting on water columns that are made to oscillate *along* the sloping bottom
(Zimmerman, 1978; Loder, 1980; Pingree and Maddock, 1985; Maas et al., 1987). In this case the direction of the vorticity flux will depend on the orientation of the tidal ellipses relative to topography, but the end result will also be time-mean currents around island and banks.

Indication of large dipole vortices associated with tidal currents have been observed in satellite images from Mokstraumen (see e.g. Figure 2), indicating that at least tidal pumping may be of importance in the Lofoten-Vesterålen region. The rectifi-
cation of tidal currents has not, to our knowledge, been observed or studied before in this region. But strong tidal oscillations around the islands of Mosken, Værøy and Røst off the southern tip of the archipelago suggest that this is a process worth investigating. In the presence of interactions with smaller-scale non-conservative flow dynamics, such time-mean circulation





cells may very well act as 'gears' that transport cod eggs and larvae, as well as nutrients and pollutants, between Vestfjorden and the outer shelf.

In this paper we will isolate these two potential transport mechanisms by conducting and analysing a purely tidally-forced numerical simulation of the region. Modeling non-linear tidal dynamics in such a complex region is challenging. Lynge et al. (2010) found that modelled tidally-driven transport through Moskstraumen is highly dependent on the model grid resolution and that a horizontal resolution down to 50–100 m is required to resolve key non-linear dynamics and thus obtain realistic transport estimates. This resolution in much higher than what is typically used in e.g. operational transport models of the

region. Our approach to this practical problem is to use an unstructured grid model which allows very high resolution in straits where nonlinear tidal dynamics is thought to be important. At the same time the flexible mesh allows us to reduce resolution away from complex geometry, thus enabling us to run simulations over a large enough domain to provide a good representation of the northward propagating tidal waves. The model setup and a validation against available observations are summarized in section 2. The two dynamical processes are then discussed separately in section 3. Finally, a brief summary of results in section

4 wraps up the study.

## 2    Model description

We use the Finite Volume Community Ocean Model (FVCOM Chen et al., 2003), for modelling tidal flows in the Lofoten-Vesterålen region. FVCOM is a prognostic, free-surface, three-dimensional primitive equation ocean model which solves the integral form of the equations on an unstructured triangular horizontal grid and a terrain-following vertical grid. For this study

of lateral transport dynamics we used a two-dimensional version of FVCOM, leaving out buoyancy effects. The model calculates momentum advection using a second-order accuracy flux scheme (Chen et al., 2013; Kobayashi et al., 1999), horizontal diffusion of momentum by the Smagorinsky closure scheme (Smagorinsky, 1963) and quadratic bottom friction using a depth-dependent drag coefficient. The governing equations are integrated in time using a modified explicit forth-order Runga-Kutta time stepping scheme (Chen et al., 2013).

The model domain, with coastline and bottom depths, is shown in Figure 3. The unstructured triangular grid enables us both to resolve small-scale nonlinear flow dynamics near land as well as the large-scale behavior of the tidal waves. Along the coast the grid resolution is as high as 30–50 meters, which provides us with a minimum of five grid cells across the narrowest cross-sections inside straits and inlets. Most straits are, however, resolved with more that five grid cells, as illustrated by Nappstraumen in the right panel of Figure 3. Such high resolution near land allows us to model flow separation and the

development of eddies, which are important processes for generating nonlinear tidal transports. The grid resolution decreases monotonically away from land and steep topography, down to around 5 km along the open boundary away from the coastline.

    Along the open boundary, we force the model with prescribed sea surface height (SSH) anomalies due to northward-propagating tidal waves. We obtain the SSH forcing fields from the TPXO 7.2 assimilated tidal model (Egbert and Erofeeva, 2002) from which we include all major constituents. The surface elevation is specified at the boundary nodes. Velocities in FV-

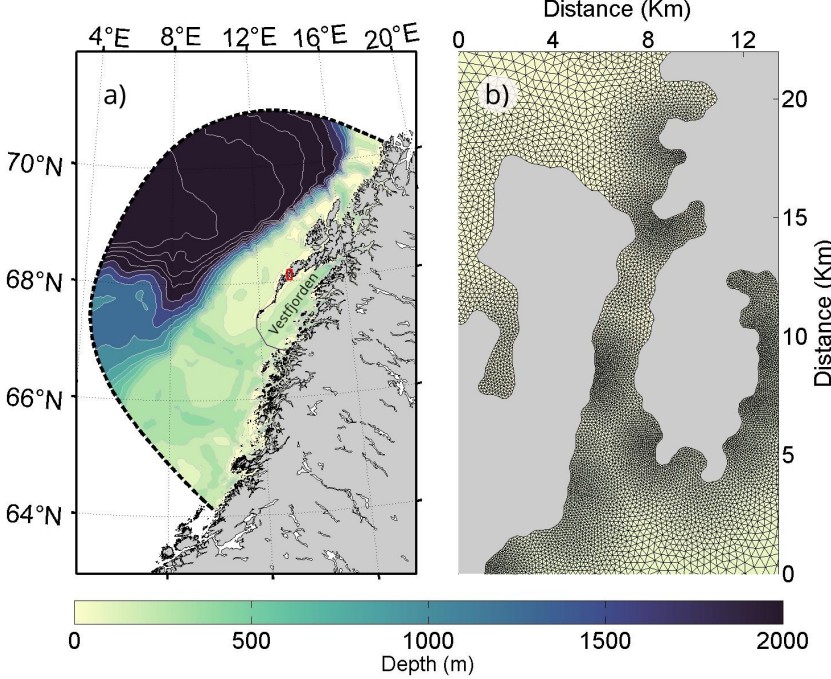

**Figure 3.** The model domain for the unstructured-grid modeling. The left panel a) shows the bathymetry inside the model domain. The dotted thick black line shows the outer open boundary of the model. The thin black line bordering Vestfjorden outlines the boundary of the region where we release a tracer. The right panel b) shows an example of the varying triangular grid resolution near Nappstraumen, highlighted by the red rectangle in the left panel.

COM are calculated in the center of each triangular cell, and not directly at the boundary. The velocities in the open boundary cells are calculated based on the assumption of mass conservation (Chen et al., 2003, 2011).

We spin the dynamics of FVCOM up for six months before analyzing the model fields. In order to investigate tidal transport dynamics, we couple FVCOM with a passive tracer module, the Framework for Aquatic Biogeochemical Models (FABM Bruggeman and Bolding, 2014). After the six-month spin-up period, we release a passive tracer of concentration $1\,\mathrm{m}^{-3}$ inside

Vestfjorden (bounded by the thin black lines shown in the left panel of Figure 3). The tracer concentration is set to zero outside. After this initial tracer release, we run the coupled model for another two more months to ensure that we capture effects of the the spring-neap cycle.

## 2.1 Model validation

The large-scale behavior of the M2 and K1 tidal waves and associated currents are shown in Figure 4. The semi-diurnal

M2 wave (left panels) is the dominating constituent in the region. The wave is scattered and deflected around the Lofoten archipelago. The fraction of the wave that enters Vestfjorden slows down and the SSH amplitude increases towards the head





of the fjord due to the geometry of the fjord. In contrast, the fraction of the wave that passes west of the archipelago speeds up along the narrowing shelf. The result is a small phase shift and a large difference in SSH amplitude between Vestfjorden and the outer side of the archipelago. This generates strong tidal currents in the straits (lower left panel). Particularly strong
currents are found over the narrow and shallow ridge south of Lofotodden.

The K1 wave is the dominating diurnal constituent (right panels), but its amplitude in SSH is only about one tenth of the M2 amplitude. The K1 wave behaves similarly to the M2 wave inside Vestfjorden, and a gradient in SSH across the archipelago produces strong diurnal tidal currents as well through the straits (lower right panel). Interestingly, along the narrow outer shelf vest of the archipelago we observe that the K1 tidal current amplitude increases northward, particularly west of Vesterålen.
For comparison, the M2 tidal current amplitude decreases in the same area. This prominent amplification of the diurnal tidal component, K1, has been attributed to the generation of diurnal continental shelf waves by Ommundsen and Gjevik (2000) and Moe et al. (2002).

The large-scale behavior of both M2 and K1 waves in our model corresponds well with results reported earlier by Gjevik et al. (1997) and Moe et al. (2002). Furthermore, the sea surface height and phase from the model fit reasonably well with
observations from five stations provided by the Norwegian Mapping Authority, Hydrographic Service (2021), as shown in Figure 5. One notable exception is the phase of the K1 tide which is too small in the model compared to observations from Andenes, Kabelvåg and Harstad. The modeled tidal current amplitudes also agree well with observations (also shown in Figure 5c). Here we also observe that the K1 tidal current dominates in station 8, Sortlandssundet, which corresponds to the enhanced current velocities for the diurnal K1 tide in Vesterålen seen in the lower right panel of Figure 4. In general, we find
that the overall performance of our FVCOM tidal simulation is acceptable, providing a good foundation for investigating tidal transport dynamics in the region.

## 3   Tidally-driven tracer transport in Lofoten

Figure 6 shows a three-day average of the tracer concentration near the end of the simulation period. We observe a pronounced net tracer exchange between Vestfjorden and the shelf outside, particularly south of Lofotodden. Water with tracer concen-
tration exceeding $0.3\,\mathrm{m}^{-3}$ is transported tens of kilometers westward on the outer shelf from this southernmost region. We also observe notable tracer transports through the longer straits Nappstraumen (4) and Gimsøystraumen (5) somewhat further north. In contrast, only a very small amount of tracer appears to be transported through the long and narrow Raftsundet (6) and Tjeldsundet (7) even further to the northeast.

A visual comparison with Figure 4 suggest that the transport scales roughly with the intensity of tidal currents, but here
we will have a closer look at the actual dynamics at play. As outlined above, the focus will be on two processes. The first is essentially a Reynolds 'pumping' of a passive tracer through straits, stemming from a correlation between fluctuations in the tidal velocity and fluctuations in tracer concentration. The the second is the generation of rectified currents around islands. We set out to clarify and summarize key theoretical aspects of each process as well as check their applicability in Lofoten-Vesterålen.





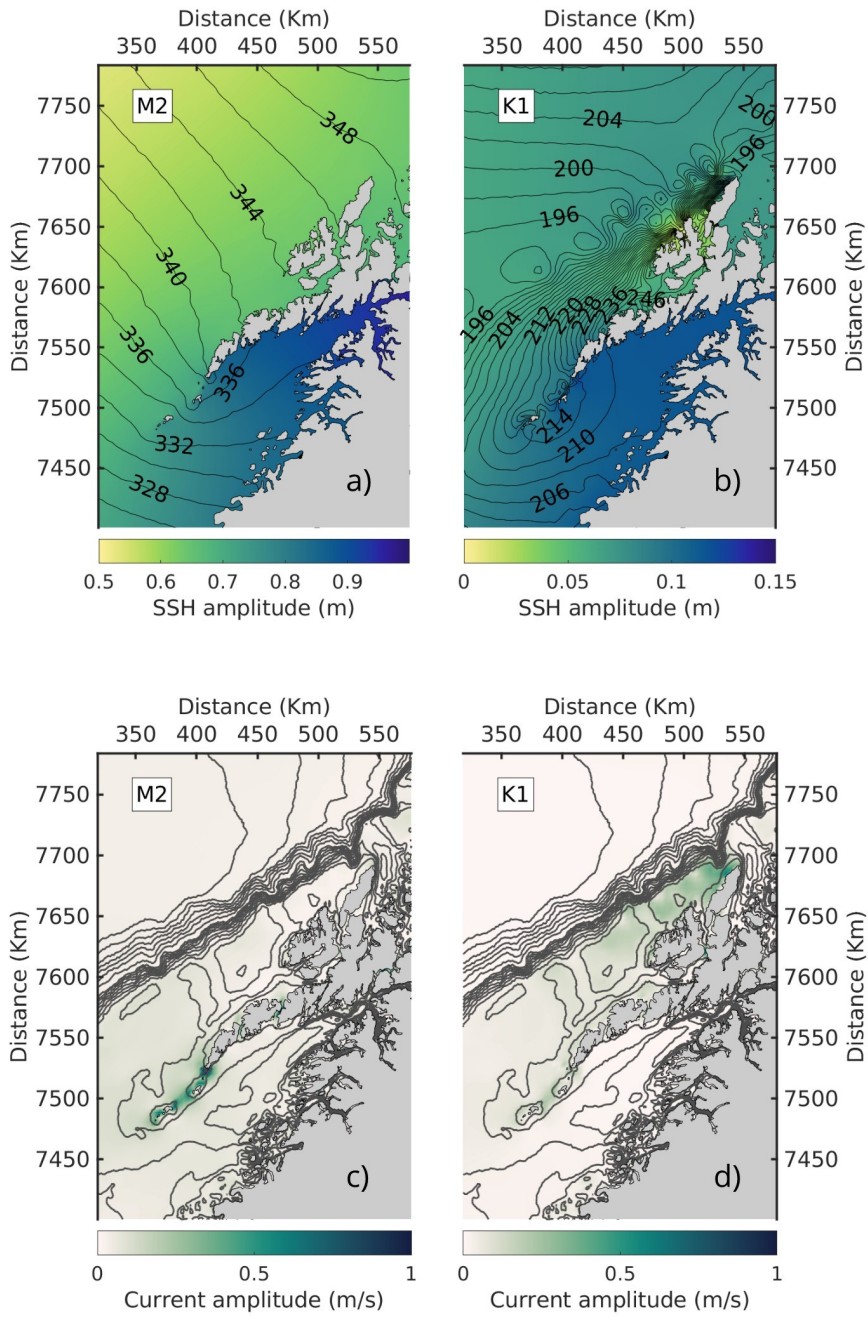

**Figure 4.** The M2 (left panels) and K1 (right panels) tide in the model. The upper panels show the amplitude (color) and phase (contours) of SSH for the two constituents. The lower panels show the magnitude of the major axis of tidal currents (colors) and bottom topography (contours).




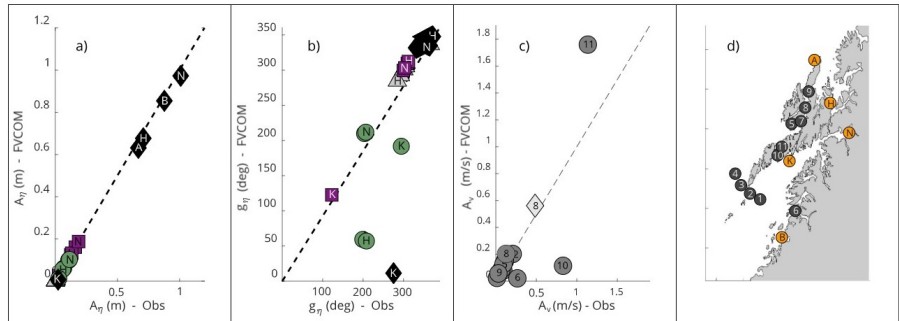

**Figure 5.** Comparison between modelled and observed tidal properties in Lofoten. Comparisons for the SSH tidal amplitude $A_\eta$ and phase shift $g_\eta$ are displayed in panel a) and in panel b), respectively, for five stations in Lofoten: Andenes (A), Harstad (H), Kabelvåg (K), Narvik (N) and Bodø (B), shown as orange markers in the right panel d). The different tidal constituents considered are M2 (black diamonds), K1 (green circles), N2 (purple squares) and S2 (gray triangles). The observations of SSH are collected from the Norwegian Mapping Authority, Hydrographic Service (2021). Panel c) shows the comparison between tidal current amplitude in the model and from observations collected from table 3 of Moe et al. (2002). In total we compare 11 stations in the Lofoten-Vesterålen region, shown as dark gray markers in the right panel d). We display the M2 tidal current amplitude from all stations, and in addition the K1 tidal current amplitude from station 8 (diamond) in Sortlandssundet, since this latter station is in a region where the diurnal tidal current (K1) is known to dominate.

## 3.1 Tidal pumping

Tidal pumping through a strait is a property exchange associated with zero net mass transport (i.e. a Reynolds flux) caused by a temporal asymmetric flow field between the ebb and the flood tide (Stommel and Farmer, 1952). The flow asymmetry arises where inflow to a strait takes the form of a broad potential flow whereas outflow is concentrated in a jet generated after flow separation. When the tidal current exits a strait, the flow decelerates as the cross-sectional area increases. If the deceleration is rapid enough for nonlinear dynamics to dominate, there will be a dynamic low pressure in the strait and high pressure outside the opening. In that case both the pressure gradient and bottom friction acts against the flow direction, and currents near the coast where friction is strongest might be brought to halt and even reverse, resulting in flow separation (Kundu et al., 2016; Signell and Geyer, 1991). Flow separation and corresponding flow asymmetry are typically present in straits that have strong tidal currents and abrupt openings. As also pointed out in the introduction, the generation of a tidal jet on outflow through an abrupt strait opening is intimately tied to the presence of a self-propagating dipole.

Before making quantitative estimates we take a look at the flow field in two of the straits. Figure 7 shows the flow and tracer field in Nappstraumen (4) through one tidal cycle. The various panels give time slices at 3 and 4.5 hours into the flood after slack tide and 3 and 4.5 hours into the ebb after the next slack tide (which corresponds to 9 and 10.5 hours after the first slack tide). At 3 hours we see that the northward-flowing tidal current has separated from the coast near the abrupt opening in the north. The separation has created two oppositely-signed vortices that are trailed by a jet, in line with previous studies (Afanasyev, 2006; Nøst and Børve, 2021). The vortices form a self-propagating dipole pair and grow in time, as can be seen at



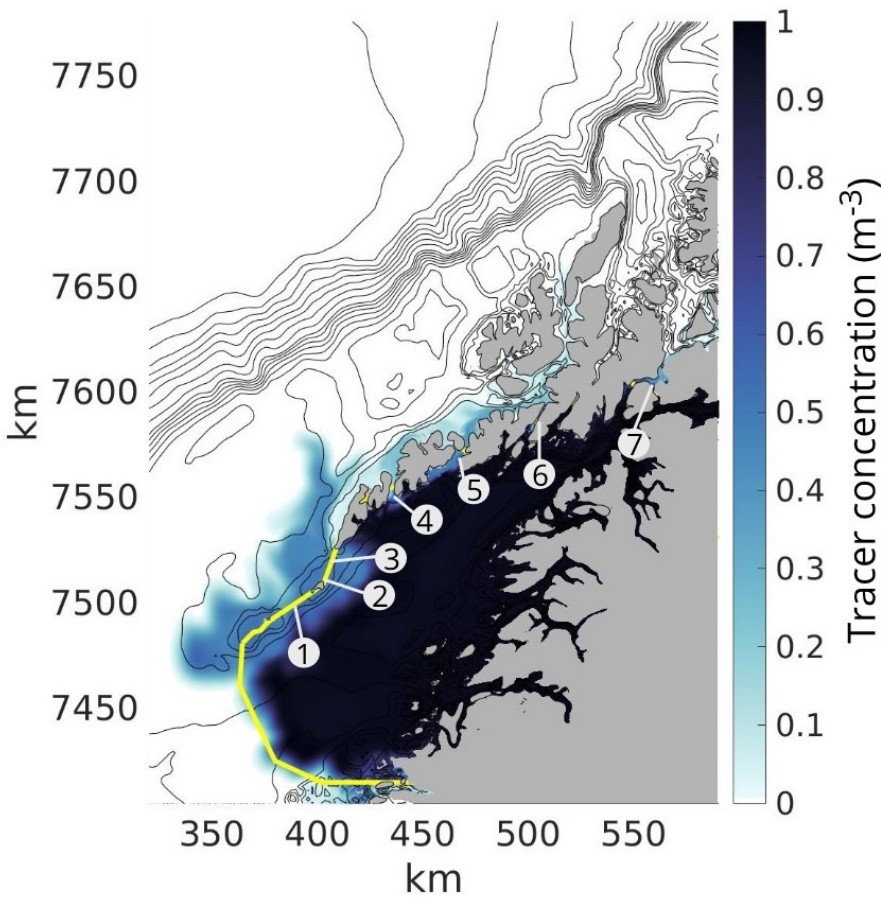

**Figure 6.** 72-hour average tracer concentration, two months after initial tracer release. The yellow line shows the boundary of the initial tracer release area. Inside the yellow boundary the initial tracer concentration was one, while everywhere else the tracer concentration was zero. The contours show the bottom topography.The main straits through the archipelago which will be investigated in this study are: (1) Røsthavet, (2) Nordlandsflaget, (3) Mokstraumen, (4) Nappstraumen, (5) Gimsøystraumen, (6) Raftsundet and (7) Tjeldsundet. Note that the numbering do not correspond to the numbering of the stations given in Figure 5.




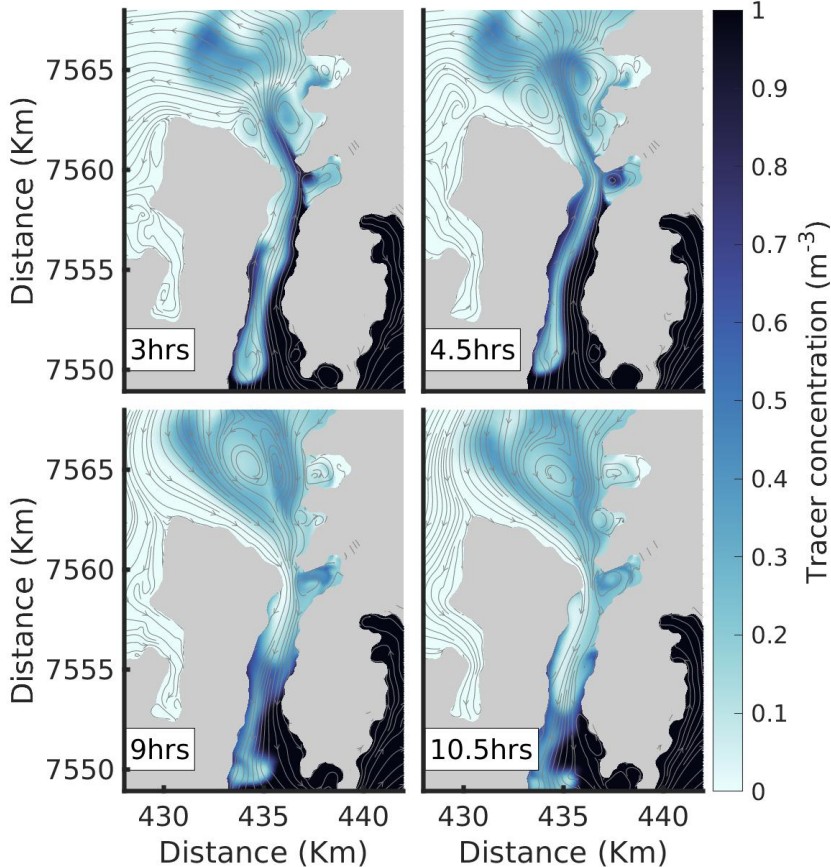

**Figure 7.** Tracer distribution in Nappstraumen (4) during the first full tidal cycle in the simulation. The time is given in hours after slack tide after ebb.

4.5 hours. The vortices clearly capture and transport waters with high tracer concentration northward as they propagate away from the strait during flood tide, as expected from theory.

The ebb tide (9 and 10.5 hours in the figure) returns water to the northern opening as potential flow, following the shape
of the coastline. The flow paths are thus distinctly different compared to those during flood tide, and waters with low tracer concentration are drawn back in, particularly along the western flanks of the strait. In this particular strait the self-propagating dipole, formed during flood, is strong enough to escape the return flow. The bulk of the tracer captured by the two vortices therefore remains at the northern side, contributing significantly to the net tracer transport through Nappstraumen over the course of the full tidal cycle. At the more gradual southern opening of the strait, there is much less indication of flow separation.
There is suggestion of a small and weak vortex pair forming along the south-western flank, but the net tracer transport appears to be limited.





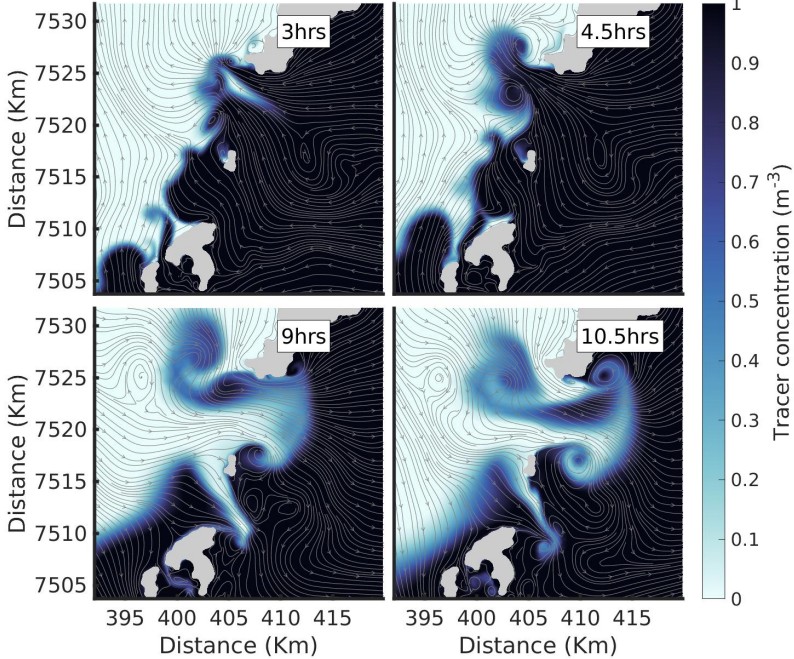

**Figure 8.** Same as Figure 7, but for Mokstraumen (3) and Nordlandsflaget (2).

The situation is somewhat different in Mokstraumen (3) between Lofotodden and the island of Mosken, as show in Figure 8. Here there is flow separation, dipole and jet formation at both exits during flood and ebb tide, respectively. A closer inspection shows that the dipoles form later in the tidal cycle compared to the generation at the northern exit in Nappstraumen, and their propagation distance is somewhat shorter when the flow reverses. Even so, their propagation speed is strong enough that the bulk of the dipoles avoid being transported back into the strait by the return flow. The inflow to Mokstraumen, in contrast, also primarily takes the form of potential flow, drawing fluid into the strait from all directions. The result is a large net tracer transport which is clearly seen in Figure 6. A dipole with a trailing jet is also observed to form during ebb tide (10.5 hours) in Nordlandsflaget (2) a few kilometers to the south-west between the islands of Mosken and Værøy. This flow feature brings low-concentration waters into Vestfjorden, but the net effect appears to be somewhat dwarfed by the pumping that takes place in Mokstraumen.

### 3.1.1 Parameters controlling tidal pumping

According to Nøst and Børve (2021), the net transport of a tracer through a tidal strait depends primarily on two non-dimensional length scales. The first parameter is a purely kinematic one, namely the ratio of the tidal excursion $L_t$ (the expected travel distance of a particle transported by the tidal current) and the length $L_{xs}$ of the strait:

$$L^* = \frac{L_t}{L_{xs}}. \tag{1}$$




If the tidal excursion is shorter than the strait ($L^* < 1$), a net transport of properties is not possible. The second non-dimensional length scale reflects the dynamics at play, namely the travel distance of the self-propagating dipole relative to the extent of the sink region:

$$L_s = \frac{L_d}{R_s},\tag{2}$$

where $L_d$ is the dipole travel distance during one half tidal period and $R_s$ is the sink radius (a measure of the region covered by potential flow on inflow to the strait). $L_s$ corresponds roughly to the nondimensional Strouhal number used by Kashiwai (1984) and Wells and van Heijst (2003). If $L_s < 1$, the dipole is inside the sink region and will be affected by the potential flow back into the strait. Depending on the self-propagation velocity of the dipole relative to the sink velocity at its positions, a smaller or larger fraction of the dipole will be pulled back into the strait.

While the first non-dimensional parameter, $L^*$, is relatively easy to estimate in our study, the second parameter, $L_s$, is more complicated to work with in a realistic setting. $L_s$ depends on the dipole properties and the shape of the sink regions, both of which are affected non-trivially by the kind of complex bathymetry and coastlines present in Lofoten. Therefore, instead of tracking dipole travel distances and estimating sink radii, we here chose to assess the flow asymmetry at the strait openings. In other words, we set out to investigate the extent to which the inflow through a strait opening behaves as potential flow whereas the outflow takes the form of a jet. As such, this relationship is more in line with the original model of Stommel and Farmer (1952) and follows the procedure recommended by Signell and Butman (1992).

To reiterate, the formation of a tidal jet during outflow from a strait requires flow separation which is driven, in part, by the build-up of an adverse pressure gradient. The build-up of an adverse pressure gradient, in turn, requires nonlinear advection of momentum (Signell and Geyer, 1991). So it makes sense to investigate the relationship between non-linearity and flow asymmetry in the various straits in Lofoten. In a coordinate system where the x-axis points along the strait, a truncated form of the along-strait momentum equation is

$$\frac{\partial u}{\partial t} + u\frac{\partial u}{\partial x} = -g\frac{\partial \eta}{\partial x},\tag{3}$$

where $u$ is the along-strait velocity, $\eta$ is the sea surface height and $g$ is the gravitational acceleration. We have ignored cross-strait advection and friction for the arguments to follow (skin friction in our simulations is demonstrably small compared to the time acceleration at tidal frequencies) . An assessment of the importance of non-linearity in a strait opening can be done by comparing the advection term to the time rate of change of momentum. The advection term itself can be estimated from volume conservation as

$$u\frac{\partial u}{\partial x} \sim u_i\frac{u_i}{\Delta x}\left(\frac{A_i}{A_e} - 1\right),\tag{4}$$

where $u_i$ is the velocity at the inner, narrow, part of the strait, and $A_i$ and $A_e$ are the cross-sectional areas covered by the current at the inner part of the strait and the strait exit, respectively. Finally, $\Delta x$ measures the distance over which the change in cross-sectional area takes place. If the tidal current is large and the change in cross-sectional area is large and abrupt (meaning $A_e \gg A_i$ and $\Delta x$ is small), then the nonlinear advection will be strong.





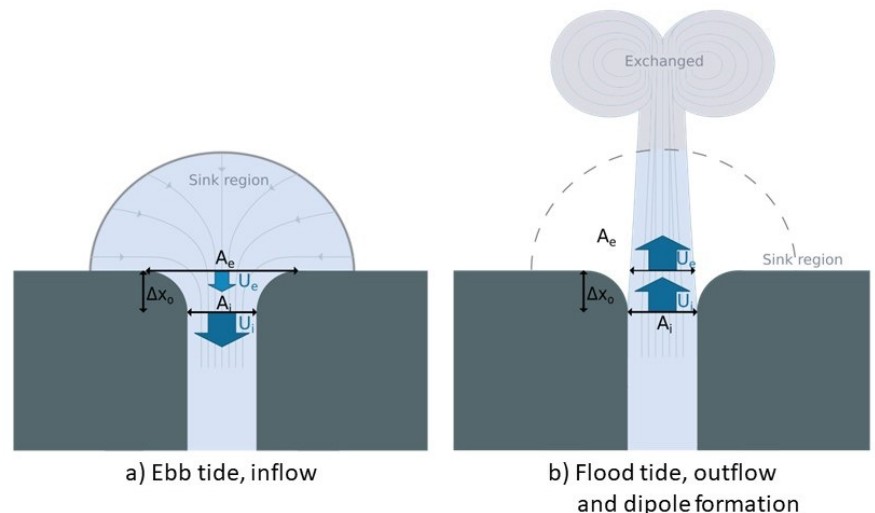

**Figure 9.** A sketch illustrating flow asymmetry. The left panel a) shows the tidal current entering the strait from all directions during ebb tide. The right panel b) show a tidal current exiting the same strait during flood tide. However, now the flow separates from the coastline and a dipole with a trailing jet has formed and propagated away from the strait. U is the tidal current speed and A is the cross-sectional area. The notations $_i$ and $_e$ corresponds to the inner and outer side of the strait opening, respectively. $\Delta x$ is the length of the strait opening where we evaluate the flow asymmetry and the nonlinearity of the flow dynamics.

The non-linearity of the flow is then found by dividing (4) by $u_i/T$, where $T$ is half a tidal period. So we get the non-linearity parameter

$$S_{nl} = \frac{u_i T}{\Delta x}\left(\frac{A_i}{A_e} - 1\right). \tag{5}$$

As shown by the sketch in Figure 9, the area covered by the jet at the strait exit, $A_e$, can be quite different between inflow and outflow. On inflow the appropriate scale for $A_e$ is the actual width of the strait exit, while on outflow the scale may be that of the jet—if a jet forms. So the *maximum* strength of non-linearity is best measured on inflow, i.e. using values of $A_i$ and $A_e$ gathered from the strait geometry.

To assess flow asymmetry, we will use the model's pressure or sea surface height field. To understand how flow asymmetry will manifest itself in the pressure field, we again return to the sketch in Figure 9. If the inflow takes the form of potential flow while the outflow is in the form of a jet (as indicated in the figure), the non-linear pressure gradient across the strait opening (i.e. over distance $\Delta x$) will be larger during inflow than during outflow. This observation suggests that the magnitude of the difference in pressure gradient between inflow and outflow will be a measure of the asymmetry.

We start by forming normalized pressure gradients across each strait openings:

$$\widetilde{\Delta \eta} = \frac{\Delta \eta_o / \Delta x_o}{\Delta \eta_s / \Delta x_s}, \tag{6}$$





where $\Delta\eta_o/\Delta x_o$ is the pressure gradient across the opening and $\Delta\eta_s/\Delta x_s$ is the corresponding gradient across the entire strait. The latter should primarily reflect the large-scale pressure gradient, so normalizing by this will help isolate the nonlinear contribution to the pressure gradients around the strait exits. The flow asymmetry around a given strait exit is then measured by the magnitude of the difference between $\widetilde{\Delta\eta}$ at flood and ebb tide:

$$A_{xo} = |\widetilde{\Delta\eta}_{flood} - \widetilde{\Delta\eta}_{ebb}| \tag{7}$$

A small value of $A_{xo}$ should indicate negligible flow asymmetry while a large value should indicate large flow asymmetry and thereby the potential for prominent tidal pumping.

We calculated $\widetilde{\Delta\eta}$ at ebb and flood tide for each M2-tidal cycle at both openings of all the straits shown in Figure 6. Individual estimates for each strait opening and each phase of the tides were then averaged over the whole simulation period. Finally, a mean asymmetry parameter $A_{xo}$ was calculated for each opening. Since we deal with realistic geometries, the definition of the openings is somewhat subjective. But we tried to apply similar criteria to all strait openings, choosing the most obvious outer strait entrance/exit and the corresponding closest narrow cross section inside. The outer opening would typically be where flow separation and dipole formation could potentially occur and contribute to tidal pumping. An example is the northern exit of Nappstraumen, which is defined to start at the narrow cross-section where the flow separates and dipole forms (see Figure 7). Corresponding nonlinearity parameters $S_{nl}$ were also estimated over the same openings for each M2 tidal cycle and averaged over these.

The estimates of $A_{xo}$ and $S_{nl}$ for the seven straits are shown in Figure 10. The calculation reveals considerable scatter but indicates a near-linear relation between the two parameters. This suggests that most straits that have nonlinear flow dynamics also have a flow asymmetry that may be linked to formation of tidal jets. We made estimates for both openings of each strait since the geometries on the two sides may be widely different. Nappstraumen (4) is the most notable example. At its northern opening, the flood exit, abrupt changes in the coastal geometry causes the flow dynamics to be highly nonlinear and asymmetric between flood and ebb. And we saw from Figure 7 that the asymmetry here is closely tied to prominent dipole formation during flood tide. In contrast, at the more gradual opening in the south, non-linearity, dipole formation and asymmetry are much weaker.

The largest nonlinearities and asymmetries are found in the northern opening of Nappstraumen (4), in both openings of Moskstraumen (3) and in the southern (ebb) opening of Nordlandsflaget (2). It is interesting to note that the non-linearity in Røsthavet (1) is comparable to that in the northern (flood) opening of Nordlandsflaget, but that the asymmetry is lower. As it turns out, Røsthavet is the widest strait in the whole region. So although tidal currents are just as large as in Nordlandsflaget and there is actually flow separation here during both phases of the tide (not shown), the vortices formed are too far apart to form a self-propagating dipole and a trailing tidal jet. The longer straits in the north (5–7) all have moderate to low nonlinearities and asymmetries. The reason for this is probably that the overall flow dynamics becomes more linear as the strait length increases (Nøst and Børve, 2021). This brings down the current speeds, and hence the nonlinearity, in these long straits.


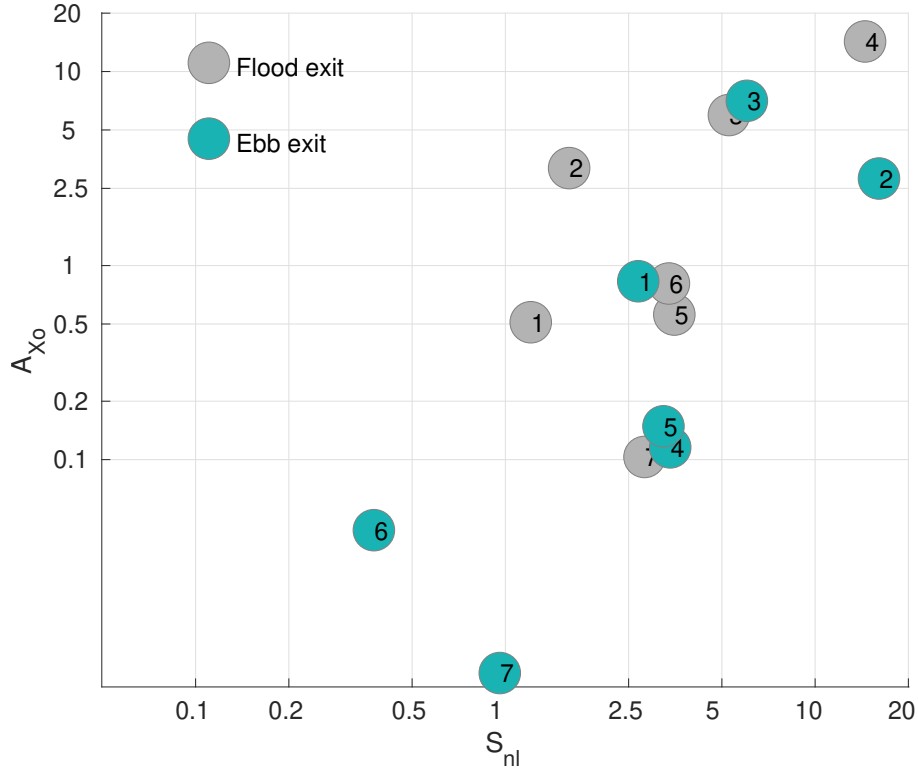

**Figure 10.** Estimates of the flow asymmetry $A_{xo}$ at the openings of each strait plotted against the non-linearity parameter $S_{nl}$. Green dots are values at the flood exit (directed out of Vestfjorden) while light gray dots are values at the ebb exits (directed into Vestfjorden). Both parameters are plotted on log scales.

### Measuring tidal pumping strength

To finally evaluate the strength of the tidal pumping, we calculate a tracer transport efficiency for each strait. The transport efficiency $T_p^*$ is defined as the actual tracer transport through the strait divided by a 'transport potential' made up of the time-averaged magnitude of the along-strait velocity $|u|$, the time-averaged mean tracer concentration difference between the two strait openings $\Delta c$ and the strait cross-sectional area $A$. So

$$T_p^* = \frac{\iint \overline{u'c'}\, dA}{\Delta c\, \overline{|u|}\, A}. \tag{8}$$

where overbars indicate the time mean and primes indicate perturbations from that mean, so that $\overline{u'c'}$ is the Reynolds flux of $c$. The transport efficiency for a given strait is estimated in the same manner as the non-linearity and flow asymmetry, i.e. by calculating a value for each M2 tidal cycle and then averaging over the whole simulation period.

Figure 11 shows $T_p^*$ for all straits plotted against asymmetry parameter $A_{xo}$ and the nondimensional tidal excursion $L^*$. The asymmetry parameter for a given strait is the average from the two strait openings. As already discussed, and as seen in panel





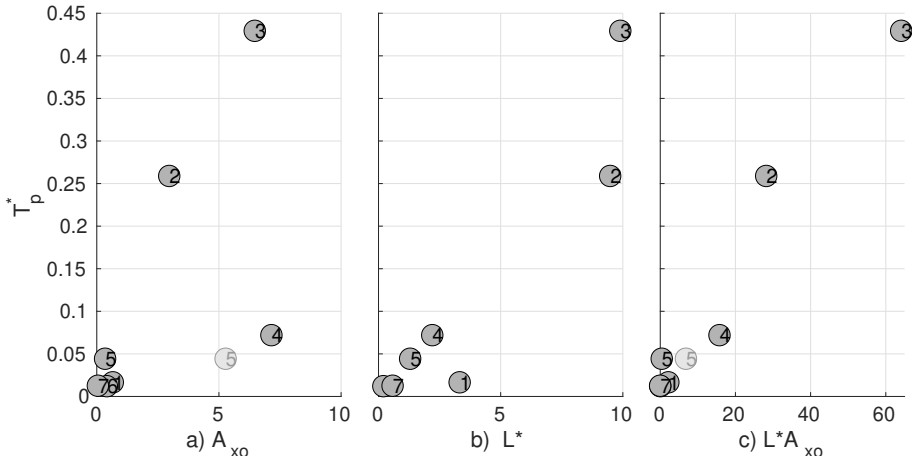

**Figure 11.** The tracer transport efficiency $T_p^*$ plotted against non-dimensional parameters $A_{xo}$ (a), $L^*$ (b) and $A_{xo}L^*$ (c). Two estimates of $A_{xo}$ are shown for Gimsøystraumen (5).

(a), three straits stand out in terms of flow asymmetry: Nordlandsflaget (2), Moskstraumen (3) and Nappstraumen (4) (where the high value comes from the northern opening). We now see that these are also the three straits with the highest transport efficiency. But even though Nappstraumen has the largest flow asymmetry of all straits, the transport efficiency is notably lower than in Nordlandsflaget and Moskstraumen. The likely reason is tied to the fact that Nappstraumen is a relatively long

strait, as can be seen in panel (b). The tidal excursion in Nappstraumen (4) is only twice the strait length, while the excursion in Nordlandsflaget (2) and Moskstraumen (3) is almost ten times longer than the strait length. Hence, just by considering the strait length, we expect the net effect of flow asymmetries in Moskstraumen and Nordlandsflaget to be larger than in Nappstraumen. Røsthavet (1) is also a short strait, where the tidal excursion is much larger than the strait length. However, in this strait the flow asymmetry is weak and we thus expect little tidal pumping. We have at present no underlying theory for tidal pumping

efficiency as a function of both $A_{xo}$ and $L^*$. But since the transport efficiency must depend on both flow asymmetry *and* short strait length compared to the tidal excursion, we plot $T_p^*$ against the product of the two parameters in panel (c). The scatter is now reduced and the data from the various straits roughly follow a linear relationship.

In forming the various estimates above some subjective decisions will impact the results. In particular, the exact value of the asymmetry parameter $A_{xo}$ depends on the location chosen for the inner and outer opening of a strait (to calculate a pressure

drop). Complex strait geometries typically make clear-cut choices difficult. Gimsøystraumen (5) is the strait which has the most complex geometry, having two regions where the strait widens in the north (not shown). In Figure 11 we have therefore shown two estimates of $A_{xo}$ for this strait, based on pressure differences taken across these two distinct northern openings. The exercises suggest that $A_{xo}$ for this strait ranges from  0.8 to  5.5, where the latter value begins to approach the asymmetry of Nappstraumen. We take the span of values in Gimsøystraumen as an upper bound for the general uncertainty in $A_{xo}$. Raftsundet

also has a complex opening in the north, however, the length of this strait is the main limiting factor for net transports by tidal pumping, and the result will not change notably due to the nonlinearity parameter. The uncertainty for the other straits, with





simpler geometries, is lower. Given this level of uncertainty, we therefore take the above calculations as clear indication that the transport efficiency through the various straits in Lofoten-Vesterålen are closely linked to the level of flow asymmetry caused by flow separation, dipole and jet formation, and to the length of the straits relative to the tidal excursion.

## 3.2 Rectified tidal currents

The second nonlinear process to be assessed is the rectification of oscillating tidal currents around the islands off the southern tip of Lofoten. Residual tidal currents encircling banks and islands have been observed in various places around the world, like the Norfolk islands and Georges bank (Huthnance, 1973; Loder, 1980). The key process, as outlined in the introduction, appears to be net vorticity fluxes generated by vortex stretching and squeezing by oscillating tidal flow over sloping bottom topography—in the presence of some irreversibility, like bottom friction.

In Lofoten, the distortion of the northward-propagating tidal waves produces particularly strong tidal currents across the shallow ridge south of Lofotodden (Figure 4). Tidal rectification around the islands located here, Mosken, Værøy and Røst, seems likely. And indeed, a zoom in on on this region in Figure 12 reveals time-mean anticyclonic (clockwise) circulation cells around the islands. There are two distinct circulation cells, one around Røst and another around Værøy-Mosken. The circulation cells reach speeds of about 0.2–0.25 m/s, which is similar in magnitude to observed background currents in the region (Mork, 1981). In Mokstraumen, the model's mean current speeds exceed 0.5 m/s, but the strongest flow here is associated with a rectified anticyclone on the inside of that strait—an anticyclone we will return to later. Figure 12 also shows the time-mean tracer field, revealing that the circulation cells advect low-concentration waters into Vestfjorden northeast of the island groups and high-concentration waters out of the fjord on the southwest sides. Much of the net tracer transport is clearly associated with the tidal pumping mechanism investigated above, but transport of tracer out of Vestfjorden south of Røst is clearly mainly tied to the anticyclonic flow around this island.

### 3.2.1 Vorticity flux and residual currents

Before doing a quantitative analysis of these currents, we will review some of the relevant theory. One useful starting point (following e.g. Zimmerman, 1978, 1981) is the vorticity balance derived from the shallow-water equations:

$$\frac{\partial \xi}{\partial t} + \nabla \cdot \boldsymbol{u}(f + \xi) = -\nabla \times \left(\frac{\boldsymbol{\tau}_b}{H}\right), \tag{9}$$

where $\xi = \nabla \times \boldsymbol{u}$ is relative vorticity, $f$ is the Coriolis parameter, $\boldsymbol{\tau}_b$ is a bottom stress and $H$ is the water depth. We have neglected forcing by a surface wind stress and also, for simplicity, lateral viscosity. In the simplified treatment below we will also only consider linear bottom friction, so that $\boldsymbol{\tau}_b = R\boldsymbol{u}$. Finally, we will ignore the sea surface height contribution to the water column thickness, i.e. apply the rigid lid approximation. Integration of (9) over the area bounded by a closed depth contour $s$, followed by the use of Stokes' theorem, gives

$$\frac{d}{dt} \oint \boldsymbol{u} \cdot \hat{\boldsymbol{t}} \, ds + \oint \boldsymbol{u}(f + \xi) \cdot \hat{\boldsymbol{n}} \, ds = -\frac{1}{H} \oint R\boldsymbol{u} \cdot \hat{\boldsymbol{t}} \, ds, \tag{10}$$




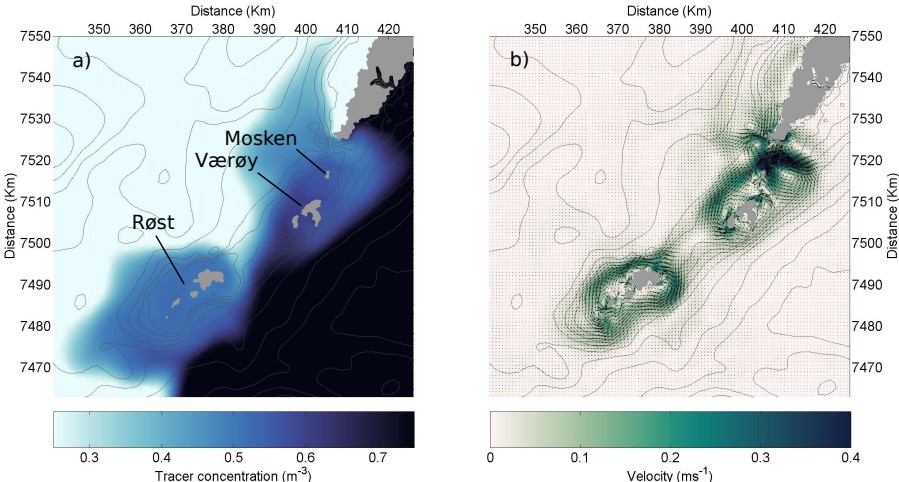

**Figure 12.** Time-mean tracer concentration (a) and time-mean currents (b) around the southern tip of Lofoten near the end of the simulation. Thin contours show the bottom topography.

where $\hat{t}$ and $\hat{n}$ are unit vectors tangential (positive clockwise) and normal (positive outwards) to the contour. We now apply the Reynolds decomposition to velocity and vorticity, splitting into means over a tidal cycle and perturbations from such means. If considering the Coriolis parameter to be constant (a very good assumption for the scales considered here), then a time average

over a tidal cycle gives the approximate balance

$$\frac{d}{dt} \oint \bar{\boldsymbol{u}} \cdot \hat{t}\, ds + \oint \overline{\boldsymbol{u}'\xi'} \cdot \hat{n}\, ds = -\frac{1}{H} \oint R\bar{\boldsymbol{u}} \cdot \hat{t}\, ds, \tag{11}$$

where, as before, overbars indicate the time mean and primes perturbations from that mean. We have ignored a term involving transport of mean vorticity by mean currents since this can be assumed to be small for oscillatory tidal forcing. Note that after the time averaging, the time evolution left in (11) is over scales longer than the fast tidal oscillations. So the expression states

that a net Reynolds flux of vorticity out of a closed depth contour will cause an acceleration of anticyclonic flow around the contour (at time scales shorter than $T \sim H/R$) and, eventually, a time-mean anticyclonic flow which balances the vorticity flux with bottom friction.

The total response to arbitrary forcing can be found by Fourier-transforming the above integral equation in time. The expression for each individual Fourier-component becomes

$$\oint i\omega\bar{\boldsymbol{u}} \cdot \hat{t}\, ds + \oint \overline{\boldsymbol{u}'\xi'} \cdot \hat{n}\, ds = -\oint \frac{R\boldsymbol{u}}{H} \cdot \hat{t}\, ds, \tag{12}$$

where, now, velocity and vorticity are functions of frequency rather than time. Depth $H$ is constant along a closed $s$ contour, and if we assume that $R$ is constant as well we get an expression for dynamic response of the mean circulation around the contour:

$$\oint \bar{\boldsymbol{u}} \cdot \hat{t}\, ds = -\frac{\oint \overline{\boldsymbol{u}'\xi'} \cdot \hat{n}\, ds}{R/H + i\omega}. \tag{13}$$




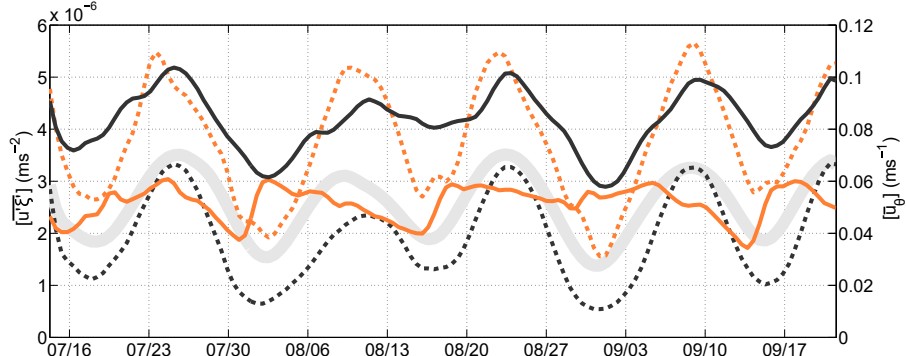

**Figure 13.** Reynolds vorticity flux ($\overline{u'\xi'}$, dashed lines) out of closed depth contours that wrap around Mosken-Værøy (orange) and around Røst (black), and azimuthal velocity ($\bar{u}_\theta = \bar{u} \cdot \hat{t}$, solid lines), both averaged around the same closed contours. All quantities have been smoothed over four M2 cycles and also averaged over a set of closed contours between 30 and 70 meters. Sea surface height fluctuations (gray thick line) over southern Lofoten are also shown.

So the prediction is a circulation whose magnitude is equal to the integrated vorticity flux scaled by $|R/H + i\omega|$ and whose phase lag is $\phi = \tan^{-1}(\omega H/R)$. The full response to forcing over a range of frequencies can then be found by solving (13) for each frequency, followed by an inverse Fourier transform. The time-dependent problem is essentially an f-plane equivalent to that of wind-driven closed-f/H variability studied by Isachsen et al. (2003), but with wind stress forcing replaced by lateral vorticity fluxes.

The primary slow time scale variation in forcing for our problem is the spring-neap cycle. So $\omega_{sn} = 2\pi/14.75\,\mathrm{rad\,days}^{-1}$. Using a typical value for bottom friction, $R = 10^{-3}\,\mathrm{m\,s}^{-1}$, and a depth $H = 50\,\mathrm{m}$, we expect a spin-up time of approximately 14 hours which corresponds to a phase lag $\phi$ of about 0.24 radians or 14 degrees. We now test these predictions on the time-mean flow cells observed around the islands near the tip of Lofoten. Figure 13 shows the Reynolds vorticity flux out of closed depth contours that wrap around Mosken–Værøy and around Røst. For each contour, a contour-averaged Reynolds flux has

been calculated for each sequential M2 tidal cycle. The resulting time series has then been low-pass filtered using a Hanning filter of width equal to four M2 cycles. Finally, for each island group (Mosken–Værøy and Røst) an average has been made over several such closed contours. The calculation clearly reveals a positive vorticity flux out of the contours (towards greater depths) at all times, and this flux is roughly in phase with the spring-neap variations in sea surface height over the region (also shown). Finally, the figure shows the low-passed azimuthal velocity (tangent to a contour) averaged around the same sets of

contours. The circulation is anticyclonic and thereby in agreement with the sign of the vorticity flux.

However, the figure also reveals that the two circulation cells respond differently to the spring-neap cycle. The cell around Røst is nearly in phase with the Reynolds flux forcing, with a phase delay of only about half a day—close to the theoretical prediction. But the flow variability around Mosken-Værøy is more erratic and, on average, lagging the forcing by 9-10 days. The amplitude of the spring-neap flow variations around Mosken-Værøy is also smaller than that around Røst even though the

amplitude of the Reynolds flux forcing is larger. Taken together, these results indicate that the theory works well at describing





the slowly-evolving anticyclonic circulation around Røst but that additional dynamics must be considered to understand the cell around Mosken-Værøy. We will return to this issue below but will first examine the underlying process that sets up the vorticity flux through these closed depth contours.

### 3.2.2 The source of the vorticity flux

The direction of the vorticity flux may be understood by following a water column that moves periodically up and down a topographic slope, driven by a large-scale tidal potential (Zimmerman, 1978, 1981). Substituting the shallow-water continuity equation into (9) gives

$$
\frac{D(f+\xi)}{Dt} = \left(\frac{f+\xi}{H}\right)\frac{DH}{Dt} - \nabla \times \left(\frac{R\boldsymbol{u}}{H}\right),
\tag{14}
$$

where $D/Dt = \partial/\partial t + \boldsymbol{u}\cdot\nabla$ is the total (Lagrangian) time rate of change experienced by the moving water column. Applying
the rigid-lid and f-plane approximations, assuming $\boldsymbol{\tau}_b = R\boldsymbol{u}$ and splitting up the friction term, gives

$$
\frac{D\xi}{Dt} = \left(\frac{f+\xi}{H}\right)\boldsymbol{u}\cdot\nabla H + \frac{R}{H^2}\boldsymbol{u}\times\nabla H - \frac{R}{H}\xi,
\tag{15}
$$

from which we can see that relative vorticity of the column has two source terms and one sink term. The first term on the RHS is vorticity production due to stretching or squeezing of the water column by flow over uneven bottom topography. If $f+\xi > 0$ motion towards deeper (shallower) water induces positive (negative) relative vorticity perturbations. The second
term is production of vorticity due to flow along a sloping bottom and often referred to as a bottom friction torque. The last term on the RHS is a loss of vorticity to bottom friction.

If we assume $|\xi| \lesssim f$ (see e.g. Table 1 of Zimmerman, 1978), then the sizes of the two production terms are

$$
\begin{aligned}
\frac{(f+\xi)}{H}\boldsymbol{u}\cdot\nabla H &\sim \frac{fUh'}{DL}, \\
\frac{R}{H^2}\boldsymbol{u}\times\nabla H &\sim \frac{RUh'}{D^2 L},
\end{aligned}
$$

where $U$ is tidal current amplitude, $D$ is mean water depth and $h'$ and $L$ are the height and length scales of the topographic feature. Comparing the magnitude of the two terms, using typical values for Mosken/Værøy and Røst ($D \sim 50\,\mathrm{m}$, $R \sim 10^{-3}\,\mathrm{m\,s^{-1}}$ and $f \sim 10^{-4}\,\mathrm{s^{-1}}$), gives $fD/R \sim 5$. This suggests that vorticity production by flow up and down topography is quite a bit larger than production by bottom friction torque. If $|\xi| > f$ the production term by squeezing and stretching of the water column becomes increasingly larger compared to production of bottom friction torque for the same depth. For simplicity we will
therefore neglect the latter term in the following, leaving the approximate expression

$$
\begin{aligned}
\frac{D\xi}{Dt} &= \left(\frac{\xi+f}{H}\right)\boldsymbol{u}\cdot\nabla H - \frac{R}{H}\xi \tag{16}\\
&= \left(\frac{\xi+f}{H}\right)\frac{DH}{Dt} - \frac{R}{H}\xi, \tag{17}
\end{aligned}
$$

or, cast in terms of potential vorticity (PV),

$$
\frac{D}{Dt}\left(\frac{f+\xi}{H}\right) = -\frac{R}{H}\xi.
\tag{18}
$$





In the absence of bottom friction, PV is conserved, and the relative vorticity of a water column will only be a function of depth (on the f-plane). So as the water column oscillates up and down a sloping bottom, it will gain just as much negative (anticyclonic) vorticity on its way up the slope as it gains positive (cyclonic) vorticity on its way down the slope. The net vorticity transport by the column across a given depth contour will therefore be zero. Crucially, friction changes this since the column will then lose some negative vorticity over shallow waters and lose some positive vorticity over deep waters. Thus, on

passing any given depth contour the column will carry an excess of positive vorticity on its way towards deep waters and an excess of negative vorticity on its way towards shallow water. The end result is a transport of positive vorticity towards deep waters. A simple sketch of the rectification process is shown in Figure 14 and a simplified mathematical model is offered in the Appendix.

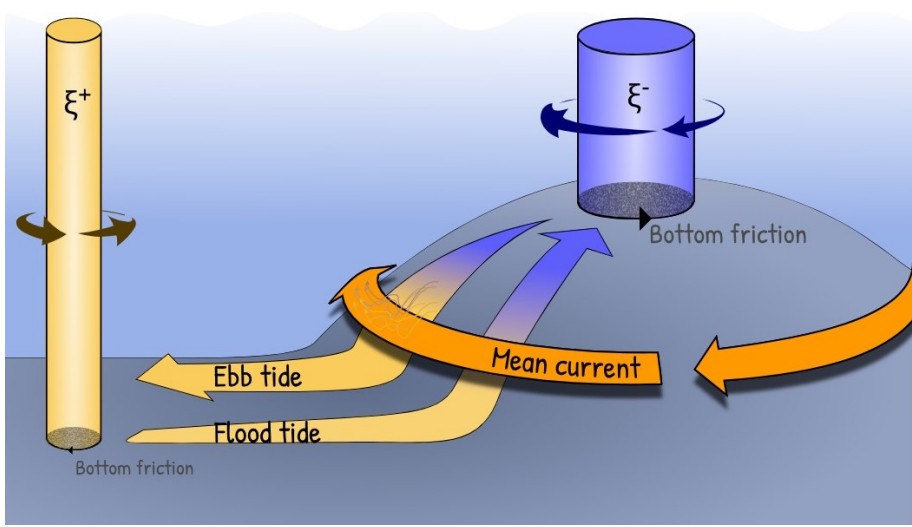

**Figure 14.** A sketch of mean-flow generation around a bank from oscillating flow across the bank topography. A water column oscillates up and down topography, attaining negative vorticity on its way up the slope and positive vorticity on its way down due to vortex squeezing and stretching, respectively. Bottom friction removes some negative vorticity from the column over shallow regions and some positive vorticity over deep regions. A sustained oscillation, by a large-scale tidal potential, will hence be associated with a positive vorticity flux from shallow to deep regions. The vorticity flux from many such columns is balanced by a mean anti-cyclonic circulation around the bank.

The net effect after integrating over the movement of many such water columns is a positive relative vorticity flux towards

deep regions. Hence, (11) predicts anticyclonic currents around a bank or island, and this is indeed what we observe in Figures 12 and 13. It is worth noting that the vorticity flux is down the large-scale background PV gradient $q_s = f/H$. So when we ignore Reynolds transport of layer thickness (in line with the rigid lid approximation), the process is qualitatively consistent with the idea of potential enstrophy dissipation via a down-gradient PV flux (Bretherton and Haidvogel, 1976; Ou, 1999).

The magnitude of the rectified current depends on the steepness of the topographic slope and the strength of the cross-slope

tidal oscillations (Zimmerman, 1978; Loder, 1980; Wright and Loder, 1985). From the scaling argument above we found that the main driver of rectification is the generation of relative vorticity by advecting water columns up and down the bottom




topography. Thus, by identifying the regions of max potential for generation of relative vorticity by cross-slope tidal currents, we can identify the areas where tidal rectification is to be expected. To look at this we ignore the effect of bottom friction, leaving

$$\frac{D\xi}{Dt} = \frac{f + \xi}{H}\frac{DH}{Dt}. \tag{19}$$

Hence, the relative vorticity change $\xi'$ experienced by a water column forced across variable topography scales as

$$\xi' = \frac{f + \xi_0}{H_0}h', \tag{20}$$

where $\xi_0$ and $H_0$ are initial vorticity and depth, respectively, and $h'$ is the topographic variation. If we assume a constant bottom slope $\alpha$, then $h' = \alpha L$ where $L$ is the lateral excursion of the water column. A topographic length scale can then be defined

as that which gives a depth excursion equal to the initial depth, or $H_0 = \alpha L_B$. By (20), such an excursion would produce the maximum relative vorticity deviation and hence the maximum potential for rectified currents. The *actual* lateral excursion experienced by parcels is given by the tidal excursion $L_T = \int \boldsymbol{u} \cdot \hat{\boldsymbol{n}}\,dt$ where, again, $\hat{\boldsymbol{n}}$ points down the topographic gradient and where the integral is taken over half a tidal cycle. Thus, $h' = \alpha L_T$. If $L_T \ll L_B$ then vorticity chances will be small since the full potential for stretching/compression is not utilized. And if $L_T \gg L_B$ then the net vorticity changes integrated over

half a tidal cycle will likely also be small due to sign reversals as the column is advected up and down topographic 'bumps'. Intuitively then, and as verified numerically by Zimmerman (1978), one expects that the largest potential for the generation of rectified currents where $L_T \sim L_B$ (see also Loder, 1980; Polton, 2015).

The ratio between time-mean $L_T$ and $L_B$ off the tip of Lofoten is shown in Figure 15. The topographic scale $L_B$ has been calculated from bathymetric data and the tidal excursion $L_T$ has been estimated using the mean M2 tidal current amplitudes

across topography. The figure also shows the time-mean flow, and there is clear indication that the rectified currents around Mosken-Værøy and around Røst are most pronounced where $L_B/L_T \gtrsim 1$. We take this as supportive evidence that the rectified currents around these islands are driven by oscillating flows over topography subject to weak bottom friction. Figure 16 shows the strength of the rectified currents around the above-studied closed H-contours as a function of $L_T/L_B$, where $L_T$ is now allowed to vary as a function of time (i.e. with the spring-neap cycle). Around Røst residual currents attain a maximum for

$L_T/L_B \sim 1.75$, with declining strengths for both smaller and larger values of the ratio. This is in agreement with theory. In contrast, the plot does not show any optimal value of $L_T/L_B$ for the flow around Mosken-Værøy. The residual current strength here instead decreases monotonically with larger values of the ratio. As we will see next, the reason for the anomalous behavior around these islands turns out to be finite-amplitude non-linear effects.

### 3.2.3   Non-linear dynamics around Mosken-Værøy

The sign of the residual currents around Mosken-Værøy is in agreement with the sign of the Reynolds vorticity flux across the closed depth contours there. But, as seen above, the time variability does not correlate trivially with the spring-neap variations in the vorticity flux. So additional dynamically processes must be at play here and, as indicated by Figure 12, a semi-persistent anticyclone southeast of Mokstraumen is likely the culprit. During each ebb tide, when the flow entering Vestfjorden through




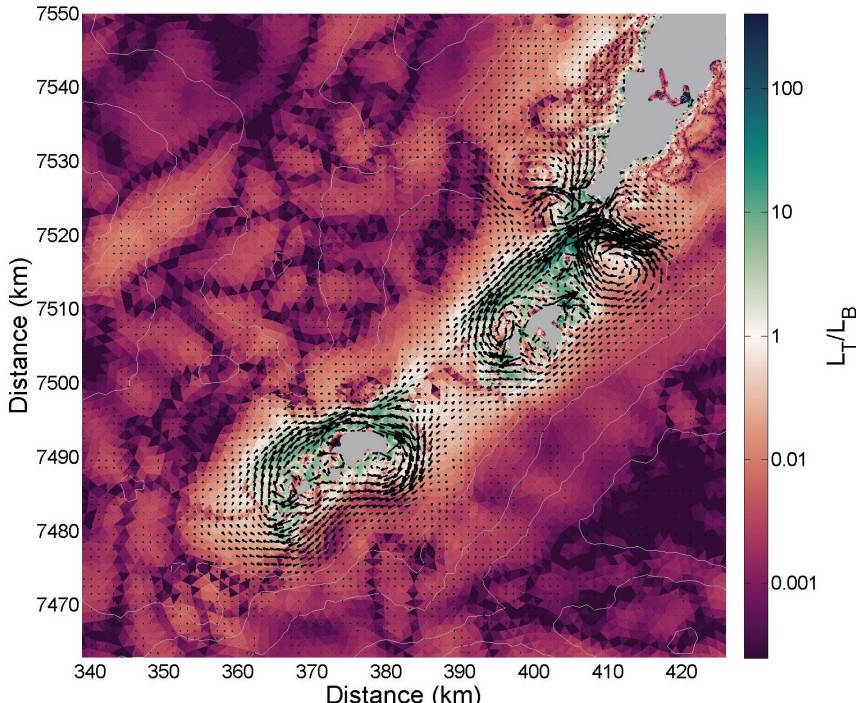

**Figure 15.** The ratio $L_T/L_B$ around southern Lofoten. Arrows show the time-mean rectified flow while black contours show bottom topography.

Moskstraumen separates from the coastline, a dipole is formed, as seen in Figure 8. After flow reversal, the cyclonic half of

the dipole is typically drawn back into Moskstraumen whereas the anti-cyclonic vortex remains on the southeastern side of the strait. The position of the anticyclone varies somewhat over time, but it is consistently strengthened by new vortex formation during each ebb phase.

Figure 17 shows streamlines of the time-mean flow in the vicinity of Mosken-Værøy. The streamlines that wrap around these two islands generally follow depth contours. But the anticyclone east of Mosken is strong enough to break topographic steering

in the northeast. Streamlines that encircle the island group detach from topography just north of Mosken to wrap around the anticyclone. The closed depth contours around Mosken-Værøy thus pass through the southwest flank of the anticyclone, so that currents from the vortex are here in the opposite direction compared to the rectified currents along the rest of the contours.

In essence, the strong anticyclone has deformed the geostrophic contours guiding the time-mean flow, and the integral analysis of (13) needs to follow this modified path. Figure 17 shows the vorticity flux and circulation around a streamline that

wraps around the island group *and* the anticyclone. Following this modified integration path shows that the circulation cell is indeed in near-phase with the Reynolds flux forcing. The figure also shows the average azimuthal velocity integrated along

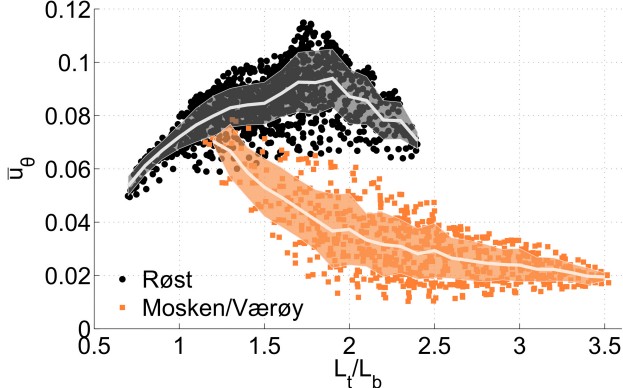

**Figure 16.** The strength of the rectified tidal currents around Røst (black dots) and around Mosken and Værøy (orange dots) are plotted against the ratio $L_T/L_B$ averaged around closed depth contours. Each dot correspond to a time mean velocity averaged over one tidal cycle. The bright thicker lines show the mean values of the residual tidal current corresponding to a given value of $L_T/L_B$ +/- 0.1. Shading indicates one standard deviation around the mean.

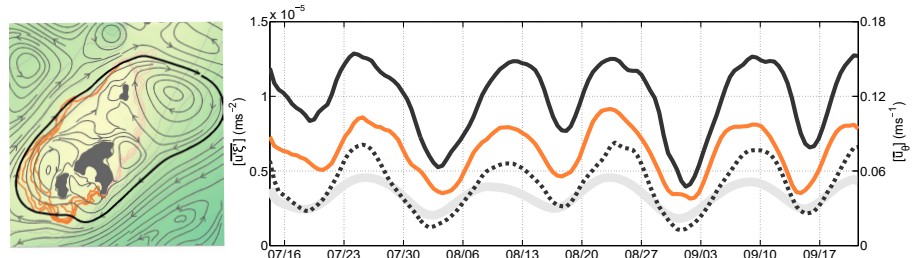

**Figure 17.** Close-up of the flow field around around Mosken-Værøy. The left panel shows time-mean streamlines (gray contours with arrow heads) as well as a set of depth contours that wrap around the island group (orange contours). The right panel shows the Reynolds vorticity flux (dashed black line) through one closed streamline which wraps around the island group and the anticylone east of Mosken (thick black contour in left panel). Also shown is the circulation around the same contour (solid black line) as well as the circulation along an incomplete depth contour south and west of the island group (orange solid line). The sea surface height variation is shown with thick light gray line.

an incomplete stretch of the original depth contours, south and west of the island group. The flow here is also in near phase with the spring-neap variations. So the circulation cell around the island group is forced by Reynolds vorticity fluxes, by the mechanism outlined above. But the strong nonlinearity in Moskstraumen makes the dynamics more complex than around the
island of Røst to the south.





## 4 Summary and conclusions

While the tides in Lofoten-Vesterålen are well known to be strong and vigorous, dominating the short-term ocean dynamics, particularly in straits and around topographic features (Gjevik et al., 1997; Moe et al., 2002), their contribution to long-term transport has gained relatively little attention. The one notable exception is Moskstraumen which is recognized as one of the main transport routes out of Vestfjorden (Ommundsen, 2002; Vikebø et al., 2007; Opdal et al., 2008; Lynge et al., 2010). Our unstructured-grid tidal simulations of the entire Lofoten-Vesterålen region confirms that Moskstraumen and, more generally, the region off the southern tip of the Lofoten archipelago is indeed the primary location for tidal dispersion in this key spawning region for the Northeast arctic cod. The main focus of this study, however, has not been quantification of transport but rather the identification of the underlying nonlinear dynamics responsible for dispersion and transport.

The flexible model grid, and the ability it offers to increase resolution in key regions, allowed us to confirm that tidal pumping, caused by flow separation and vortex dipole formation at the openings of the many straits in Lofoten-Vesterålen, is a near-ubiquitous process here. But not all straits are created equal. Strong non-linearity due to high flow speeds and abrupt strait openings, as well as short strait lengths, appears to be the explanation for why Moskstraumen and Nordlandsflaget have the highest tidal transport efficiencies in the region. The longer straits further north all have lower pumping efficiencies. But notable pumping also takes place in Nappstraumen and Gimsøystraumen.

Our simulation also revealed non-linear rectification of tidal oscillations, leading to the generation of time-mean anticyclonic circulation cells around the island groups of Mosken-Værøy and Røst off the southern tip of the archipelago. From our knowledge, tidal rectification in southern Lofoten has neither been investigated nor recognized before. But the observed rectification in our model seems to be in agreement with well-established theory of vorticity fluxes driven by cross-topographic tidal oscillations in the presence of bottom friction. The model predicted rectified current speeds up to 0.3–0.4 m/s, values that are comparable with observed background currents in this region. The circulation cell around Røst appears to be a particularly important and hitherto unknown mechanism for tracer transport around the southern tip of the archipelago.

The nonlinear tidal dynamics studied here, particularly flow separation and dipole formation, occurs on small spatial scales. In studying idealized model simulations of tidal pumping, Nøst and Børve (2021) found that a grid resolution of 50 m along the coast was necessary to realistically capture flow separation in the viscous boundary layer, but maybe not sufficient to properly resolve the vortices that form at the separation point. More specifically, the study showed that the vortices consistently formed near the smallest scale that could be resolved by that model. Lynge et al. (2010) also found that particle dispersion in realistic model simulations of Moskstraumen was highly sensitive to grid resolution and that a resolution of 50–100 m was needed for obtaining what they reported to be realistic dispersion rates. In our unstructured-grid model most of the straits had a grid resolution of 30–50 m near the coastline, so the underlying mechanisms of flow separation and vortex formation were fairly well resolved. However, due to computational constraints we had to decrease the resolution considerably away from the straits and coastlines. So since the properties and behavior of the dipoles might be influenced by grid resolution along their travel path, we expect our simulations as well to be hampered by resolution issues. Thus, we refrained from making quantitative estimates





of transport parameters like relative dispersion and lateral diffusivities, which are known to be sensitive to model resolution
(Geyer and Signell, 1992; LaCasce, 2008; Lynge et al., 2010).

Our simulations were also limited by their 2D nature. A 2D configuration was chosen to help isolate nonlinear lateral tidal dynamics, but the model was thus unable to account for baroclinic effects, e.g. the possible generation of hydraulic jumps and vertical mixing around strait openings (Lynge et al., 2010) or the establishment of density fronts around the rectification cells (Ou, 2000). In reality, baroclinic flow dynamics will impact tracer transport, both vertically and laterally. But key features of

the model's lateral flow dynamics appears to be robust. Flow separation and dipole formation in Moskstraumen, for example, is largely in agreement with observational evidence, seen e.g. in satellite data (Figure 2). The suggestion that there are anticyclonic time-mean currents around the island groups of Mosken-Værøy and Røst, generated by tidal rectification, is however worthy of a new and dedicated observational study.

Notwithstanding model limitations, the present study supports previous claims that tides are an important contributor to the

transports of Northeast Arctic cod eggs and larvae out of Vestfjorden. Even if the main transport routes due to tides coincide with transport routes following the mean flow, i.e. through Moskstraumen and south of Røst, the net transport could potentially be significantly enhanced when nonlinear tidal dynamics are present. In truth, the connectivity between the inner and outer shelf likely relies on the *interaction* between tidal dispersion and transport by the time-mean currents (Ommundsen, 2002). Additionally, our study suggests that tidal pumping through straits further north along the archipelago, in particular Napp-

straumen and Gimsøystraumen, could provide alternative transport routes to the shelf. In an on-going study, we analyze 3D unstructured-grid simulations driven by realistic atmospheric, river and lateral boundary forcing. The aim there is to investigate the relative importance of tidally-induced transport of cod eggs and larvae compared to, or in combination with, other transport processes in this region. The more realistic 3D study will hopefully also add to the general understanding of the role of nonlinear tidal dynamics in similar coastal regions, with the ultimate aim of providing more accurate transport estimates of fish eggs

and larvae, as well as pollutants, nutrients and other properties that affect the coastal ecosystem.

*Data availability.* Data is available on request

## Appendix A: A one-dimensional model of tidal rectification

We consider the Lagrangian time evolution of a water column subject to linear bottom friction:

$$\frac{D}{Dt}\left(\frac{f+\xi}{H}\right) = -\frac{1}{H}\nabla \times \frac{R\boldsymbol{u}}{H}. \tag{A1}$$

For simplicity we will assume that the RHS is dominated by velocity gradients, giving

$$\frac{D}{Dt}\left(\frac{f+\xi}{H}\right) = -\frac{R}{H^2}\xi. \tag{A2}$$

As formally laid out by e.g. (Zimmerman, 1978), we now consider the situation where the column is forced to move up and down topography by tidal currents that are dictated by remote dynamics. Thus, $H = H(t)$ is specified. The relative vorticity





$\xi$ however is assumed to be a local response to the vortex compression/stretching by this movement across topography and to
the effects of friction.

Equation (A2) can be written out to give a first-order ordinary differential equation:

$$\frac{D\xi}{Dt} - \left(\frac{1}{H}\frac{DH}{Dt} - \frac{R}{H}\right)\xi = \frac{f}{H}\frac{DH}{Dt}. \tag{A3}$$

This takes the form of a forced equation for $\xi$ with damping, where the damping coefficient is non-constant. We now define

$$p(t) = \frac{1}{H}\frac{DH}{Dt} - \frac{R}{H}, \tag{A4}$$

and multiply (A3) by $\exp\left(-\int p(t)\,dt\right) = \exp\left(\int R/H\,dt\right)/H$ before integrating in time. The expression becomes (after applying integration by parts to both sides):

$$\int_0^t \frac{D}{Dt}\left(\frac{\xi}{H}e^{\int R/H\,dt}\right)dt = -\int_0^t \frac{D}{Dt}\left(\frac{f}{H}e^{\int R/H\,dt}\right) + \int_0^t \frac{fR}{H^2}e^{\int R/H\,dt}\,dt, \tag{A5}$$

and the solution is

$$\xi(t) = \underbrace{(\xi_0 + f)\frac{H}{H_0}e^{-\int_0^t R/H\,dt}}_{\text{T-I}} - f\underbrace{\left[1 - RHe^{-\int_0^t R/H\,dt}\int_0^t \frac{1}{H^2}e^{\int_0^t R/H\,dt}\,dt\right]}_{\text{T-II}}, \tag{A6}$$

where $\xi_0$ and $H_0$ are the relative vorticity and bottom depth at $t=0$. Here terms T-I describe an exponentially-decaying
adjustment from the initial state, whereas terms T-II describe a part of the solution which achieves statistical equilibrium with
the forcing. After a few tidal cycles $\exp\left(-\int_0^t R/H\,dt\right) \to \exp\left(-rt\right)$, where $r$ is an inverse time scale for the adjustment
from initial to steady state. As seen, this spin-up time scale depends on the friction coefficient and the bottom depth variations
experienced by the column.

We now evaluate (A6) numerically for a very simplified configuration consisting of forced flow over a linear topography,
i.e. for $H = H_0 - \alpha r(t)$, where $\alpha$ is the bottom slope and $r(t)$ is the cross-slope excursion from $r(t=0)$ where $H = H_0$.
For added simplicity we assume a sinusoidal cross-slope tidal current, $v_r(t) = A\cos(\omega t)$ for tidal amplitude $A$ and frequency
$\omega$, so that $H(t)$ also becomes sinusoidal. A solution, for parameter choices $\xi_0 = 0$, $H_0 = 500\,\mathrm{m}$, $\alpha = 0.1$, $A = 0.5\,\mathrm{m\,s^{-1}}$,
$\omega = 1.4 \times 10^{-4}\,\mathrm{rad\,s^{-1}}$ (M2) and $R = 0.003\,\mathrm{m\,s^{-1}}$, is shown in Figure A1. The relative vorticity of the water column reaches
a statistically-steady state after about 10 tidal periods (about 5 days for M2 tidal forcing), this corresponding to 2–3 e-folding
scales. The column then has positive and negative relative vorticity over deep and shallow parts, respectively. The amplitude
is largest over deep parts due to the inverse dependence of depth in the friction term (see eqn. A2). Interpolating this vorticity
field to mid-depth ($H = H_0$) and taking the product of the velocity gives the Eulerian vorticity flux. The result is shown in
Figure A2 for two choices of bottom friction $R$ and five choices of initial vorticity $\xi_0$. The initial vorticity flux can be up or
down the slope, depending on $\xi_0$. But after the initial adjustment period (which depends inversely on $R$), the end result is
always a positive vorticity flux towards deep water. The magnitude of the flux is linearly proportional to $R$, as can be deduced
from (A6).



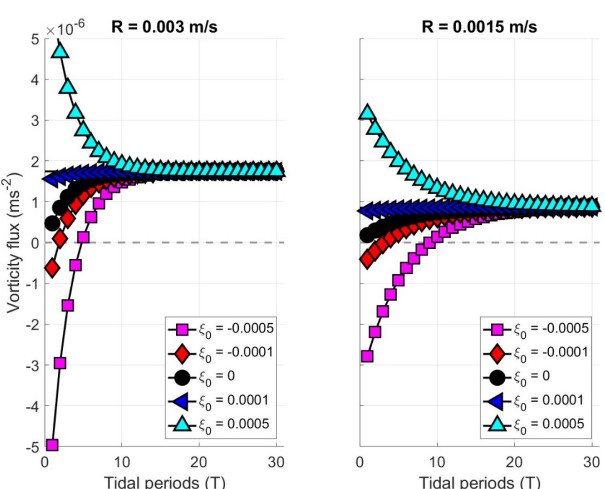

**Figure A1.** Vorticity evolution of a water column forced to oscillate over a linear bottom slope, for $\xi_0 = 0$, $H_0 = 500\,\mathrm{m}$, $\alpha = 0.1$, $A = 0.5\,\mathrm{m\,s^{-1}}$, $\omega = 1.4 \times 10^{-4}\,\mathrm{rad\,s^{-1}}$ (for M2) and $R = 0.003\,\mathrm{m\,s^{-1}}$. The red line is the transient solution (terms T-I in eqn. A6), the blue line is the statistically-steady solution (terms T-II) and the black line is the full solution.

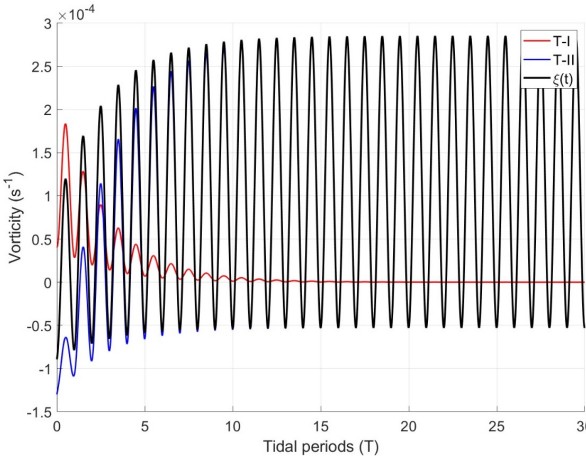

**Figure A2.** Vorticity flux averaged over an integral number of tidal period as a function of time, for the solution of (A6). Positive values indicate a vorticity flux towards deep waters. The left and right panels show results for $R = 0.003\,\mathrm{m/s}$ and $R = 0.0015\,\mathrm{m/s}$, respectively, starting from five different initial vortices $\xi_0$. The other parameters are as in Figure A1.

*Author contributions.* OAN and EB sat up the numerical model. EB conducted the numerical experiments and analyzed the data. All authors contributed in discussing and interpreting the results. EB and PEI wrote the initial paper draft and all authors have been contributing in editing the paper.

*Competing interests.* No competing interest are present

*Acknowledgements.* We thank Trygve Halsne for providing the processed satellite images of Mokstraumen and Nordlandsflaget in Figure 2. E. Børve was funded by VISTA – a basic research program in collaboration between The Norwegian Academy of Science and Letters, and Equinor (project no.6168).





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
