# Peer review of "Rectified tidal transport in Lofoten-Vesterålen, Northern Norway"

_Ocean Science, 2021_

## Author Comment (AC1)

We thank the referee for good comments and suggestions regarding our manuscript. Regarding suggestions for improving the writing and grammar, as well as figures and figure captions, we will implement them as suggested and will therefore not comment on all of these here. For more major questions and comments, we have tried to answer them one by one below.

**Introduction:**

*L.18–21: could do with a few additional references*

Good point. We will include a few additional references in the revised manuscript, both on the topic of vertical mixing and on lateral transport by non-linear dynamics (see also reply to comment below on bringing the work into a general research context in the Conclusions).

*L 50-65: It would be nice to have a diagram to illustrate this process. You could move Figure 9 to the intro and refer to it here as it greatly adds to the intuitive understanding of the process.*

*L 68-83: Again, it would be nice to have the diagram for this in the intro.*

This is an interesting suggestion. We will consider moving the two diagrams to the intro to help illustrating the two transport processes better. It may add some difficulties since these figures, especially Fig. 9, introduce some parameters that may not need to be introduced at this early stage. But we will give it a try.

**Methods:**

*L 118: Why do you use TPXO7.2? The latest version is 9 and is available as a higher resolution dataset. This is likely to affect your results. By how much do open boundary forcing values differ between the two datasets?*

The actual model run was conducted a few years back, but we got delayed in analyzing the data and writing this paper. This is why an older version of TPXO has been used. However, we believe that the validation against previous studies as well as observations is the most important check regarding model performance. We take the favorable comparisons shown in Figure 5 as indication that this model is adequate for investigating what are, first and foremost, *qualitative* aspects of the tidal dynamics and transport in the region.

**Model validation:**

*It would be nice to see some numerical values so that the reader can assess the performance of the model in absolute terms. Could you add a table with the amplitudes, phases, and amplitude and phase errors for each station for M2 and K1? You could also report mean errors for each constituent analysed.*

We will include such a table as part of the model validation.

**Tidally driven transport:**

***L.177: Why do you show results only for 3 and 4.5 hours? It would be good to see the process through the tidal cycle at e.g. hourly intervals.***

We agree that a higher temporal resolution would be useful in these figures. However, we chose to show only 3 and 4.5 hours after the two slack tides due to space constraints. Essentially, including many more time-steps will make each subfigure smaller and harder for the reader to interpret. The main purpose of these figures is to show that dipoles form, propagate away from the strait, and partly escape the return flow.

But, as said, we agree with your point. So, we will try to add a few more time-steps without compromising the quality of the figures.

**Rectified tidal transports:**

***L.335: I'm not sure this sentence fully makes sense? Which mechanism dominates and is responsible for the net flux or how does each process contribute?***

We agree with you here and will modify the text here in the revised manuscript. The main message that we will try to get across is that tidal pumping is the dominating mechanism causing net fluxes out of Vestfjorden in general. But south of Røst the net flux is mainly due to tracer advection by the rectified tidal currents (not tidal pumping). A possible modified text (also including the sentence above):

"Figure 12 also shows the time-mean tracer field, revealing that the circulation cells advect low-concentration waters into Vestfjorden northeast of the island groups and high-concentration waters out of the fjord on the southwest sides. So even though much of the net tracer transport south of Lofotodden is due the tidal pumping mechanism investigated above, there is also a contribution driven by anticyclonic mean flows around the islands here. This mechanism appears to be particularly important south of Røst where, it should be noted, there can be no formation of self-propagating dipoles."

***L343: How will your choice of linear bottom friction impact your results? Is this the same bottom friction used in the tide model? It may be an issue if they are not consistent?***

The numerical model uses quadratic bottom friction. But in our discussion of theory, we have used linear bottom friction in order to arrive at closed-form analytical expressions (eqns. 12 and 13). We will comment on this in a revised manuscript.

As for comparison between simplified theory and numerical results (with regards to the response to the spring-neap cycle), we primarily seek an order-of-magnitude agreement since the theoretical expression is obviously simplified. The reviewer's comments nevertheless led us to improve on our previous

guestimate for an effective linear drag coefficient. Essentially, we can set $R = C_d * |u|$, where $C_d$ is the model's quadratic drag coefficient and $|u|$ is an expected magnitude of current strength.

As outlined in the text, we investigated 5 and 9 closed depth contours around Mosken-Værøy and around Røst, respectively. Updating these calculations, now using the model's actual drag coefficient ($C_d$= 0.0025) and mean values for current strength $|u|$ (0.29 and 0.23 m/s for Mosken/Værøy and Røst, respectively), we found effective linear friction coefficients to be $R = 8.6 \cdot 10^{-4} \, m/s$ and $R = 5.7 \cdot 10^{-4} \, m/s$, for Mosken/Værøy and Røst, respectively. These are mean value for all the closed depth contours around each of the island groups. Using these updated values in our expression for phase lag,

$$\phi = tan^{-1}\left(\frac{\omega_{msf} H}{R}\right),$$

we obtain an updated mean phase-lag of the response to the spring-neap cycle of 0.6 days (0.24 rad/s) and 0.9 days (0.39rad/s), for Mosken-Værøy and for Røst, respectively. We will update the text with these new estimates.

**L375: How does the choice of filter impact the curves? It would be nice to see 1 tidal cycle values as well.**

The purpose of the low-pass filter is to make the figure cleaner by removing higher frequency variations (e.g. diurnal). But it's possible to add these with thinner lines. Attached here is a sample figure where fast variations have been added. We will consider using this in the revised manuscript.

[Figure]

**L440: I'm not sure I understand how you define the topographic length scale**

**L445: Where do you define $L_B$ and what is it? How does it differ from L?**

We see that we have forgotten to define the topographic length scale in the manuscript and thank the referee for pointing this out for us. We will fix this in the revised manuscript. But for now: $L_B$ is the topographic length scale, defined as follows:

$$L_B = \frac{H_0}{\alpha}, \, L_B = \frac{H_0}{\Delta H}$$

where $H_0$ is a mean depth and $\alpha = \Delta H$ is the slope of the bottom topography. $L_B$ is thus used to characterize bathymetric feature, which will cause water to move vertically over a ridge or a bump etc. at the sea floor bottom. $L$ on the other hand is the horizontal distance the water column has moved within some time dictated by a given dynamical process. In our case here, $L = L_T$, where $L_T$ is half a tidal cycle.

**Summary and conclusion:**

***L.492: 'But not all straits are created equal' – reword to something more specific.***

This was a play on words originating from the American Declaration of Independence. We may reword this general transitional sentence, but the specifics are given in the two sentences that complete that paragraph.

***General: Given that your introduction talks about cod quite a lot, the reader feels a little let down in the conclusions. How important are these tidal processes for the dispersal of the Arctic cod?***

This is a good point. Since we only investigate purely tidally-driven flows in this manuscript, we cannot say much about the contribution relative to other processes that impact the drift of cod eggs out of Vestfjorden (winds, background currents, freshwater run off etc.). We already point towards our more complete 3D follow-up study, where such a comparison will be done, at the very end of the Conclusions section. But we will downplay the cod-eggs discussion in the introduction somewhat, to not promise more than we can keep.

***How do the observed processes compare to other regions in the world and different observations? It would be nice to see this work be put into context of other previous work in this section.***

This is a useful suggestion. We will put the work a bit more into context of previous work in the revised manuscript. Below are a few examples of studies that are relevant upon a discussion in the manuscript, in which we will elaborate a bit more in the revised manuscript..

Tidal pumping, particularly in relation to tidal flushing and estuaries have been widely studied, and the formation of dipole vortices are observed many places where prominent tidal currents exit narrow straits (e.g. Aransas Pass: Whilden et al, 2014; Messina Strait: Cucco et al, 2016, Great Barrier Reef: Delandmeter, 2017). Delandmeter et al (2017) investigate dipoles formation and interaction between a line of islands which are separated by 1-2 km, which is comparable geometry to Moskstraumen and Nordlandsflaget. While the authors do not estimate the net transports through the straits, both observed and modeled behavior is qualitatively similar to what we find in southern Lofoten.

Regarding tidal rectification, George's bank is the most widely studied. Here, current meter and drifter observations have been interpreted in light of both modelling and theoretical studies (Loder, 1980; Limeburger & Beardsley, 1996; Chen et al, 2001). The residual currents encircling the bank—in a

clockwise fashion—is of similar strength as the flow we model around Røst and Mosken/Værøy, with mean Eularian speeds in the range of 0.2-0.3 m/s. By calculating the average topographic length scale and comparing it to the cross-isobath length scale of the mean current from values provided in table 1 in Loder (1980), we find that on the northwestern side and northern side $L_T/L_B \simeq 1$, equivalent to what we find for the islands of southern Lofoten. The Norfolk sandbanks are another famous example where tidally-induced circulation patterns are observed, but past studies (Huthnance, 1973; Howart & Huthnance, 1984) do not provide estimates of $L_T/L_B$ ratios for this region.

Chen, C., Beardsley, R. and Franks, P.J., 2001. A 3-D prognostic numerical model study of the Georges Bank ecosystem. Part I: physical model. *Deep Sea Research Part II: Topical Studies in Oceanography*, *48*(1-3), pp.419-456.

Cucco, A., Quattrocchi, G., Olita, A., Fazioli, L., Ribotti, A., Sinerchia, M., Tedesco, C. and Sorgente, R., 2016. Hydrodynamic modelling of coastal seas: the role of tidal dynamics in the Messina Strait, Western Mediterranean Sea. *Natural Hazards and Earth System Sciences*, *16*(7), pp.1553-1569.

Delandmeter, P., Lambrechts, J., Marmorino, G.O., Legat, V., Wolanski, E., Remacle, J.F., Chen, W. and Deleersnijder, E., 2017. Submesoscale tidal eddies in the wake of coral islands and reefs: satellite data and numerical modelling. *Ocean Dynamics*, *67*(7), pp.897-913.

Howarth, M.J. and Huthnance, J.M., 1984. Tidal and residual currents around a Norfolk sandbank. *Estuarine, Coastal and Shelf Science*, *19*(1), pp.105-117.

Huthnance, J.M., 1973. Tidal current asymmetries over the Norfolk Sandbanks. *Estuarine and Coastal Marine Science*, *1*(1), pp.89-99.

Limeburner, R. and Beardsley, R.C., 1996. Near-surface recirculation over Georges Bank. *Deep Sea Research Part II: Topical Studies in Oceanography*, *43*(7-8), pp.1547-1574.

Loder, J.W., 1980. Topographic rectification of tidal currents on the sides of Georges Bank. *Journal of Physical Oceanography*, *10*(9), pp.1399-1416.

Whilden, K.A., Socolofsky, S.A., Chang, K.A. and Irish, J.L., 2014. Using surface drifter observations to measure tidal vortices and relative diffusion at Aransas Pass, Texas. *Environmental Fluid Mechanics*, *14*(5), pp.1147-1172.

**Figures:**

Here we will only shortly comment on the direct questions. Suggestions of technical improvements and rephrasing figure captions will be implemented in the revised manuscript.

*Figure 2: It would be nice to have more information with this figure: e.g. (a) how do the two figures differ? (I guess it's the phase of the tide); (b) the velocity front is identified – why is this relevant and what does it mean for this study?*

The purpose of this figure is to show that formation dipoles and tidal jet are actually observed in Moskstraumen, and not only a result of numerical modelling. We will reformulate to bring this point up in the text and also add some more info in the caption.

*Figure 4: how about plotting observed values on top as shaded circles on top of the model results? Also, plot against latitude and longitude rather than distance.*

The suggestion of plotting observed values on top of the model results is a good idea. However, the observations are in general located near land, inside straits, where gradients are large. We are therefore weary that adding shaded circles may make the figure messy and potentially hide the overall structure of the tides. But we will consider if we can include validation here in a good way. However, we will fix these figures to plot against latitude and longitude instead of distance.

*Figure 7: Could you take your panels a bit further south? At 10.5 hours you are missing part of the incoming waters*

Yes, we will experiment with this. Note, however, that at 10.5 hours the current is southward. So extending the panels further south will include more of the outflowing water rather than incoming water (3 and 4.5 hours contain northward-directed flow).

*Figure 9: Panel b: Why is $A_e$ in the sink region? Should it not be in the jet? It would be nice to illustrate all of the length scales on this plot (as far as possible)*

$A_e$ is in the jet (the black two-headed arrow). But we see that the location of the label is poorly placed. We will fix this in the revised manuscript and also include an arrow showing the sink radius.

*Figure 13: What do you mean by 'sets of closed contours'? Could you show these on a map?*

The "sets of closed contours' should have been phrased as "sets of 5-9 closed depth contours encircling the island groups". We will include a map similar to the one in Figure 17 for the readers to see the locations of these contours.

**Minor comments:**

Thanks. All these suggestions will be implemented.

---

## Author Comment (AC2)

We thank the referee for good comments and suggestions regarding our manuscript.

First, regarding the typos in the text, we will fix all of these in the revised manuscript. Below are responses to more extensive comments and suggestions.

**Line 404: The along-isobath velocity scale U may be much stronger than the cross-isobath current used in the previous scaling (line 403).**

Good point. The amount of asymmetry between the two velocity components (along and across topography) will depend on both time and length scales of the problem. For high-frequency tidal oscillations the cross-isobath component need not be severely constrained (as it will be e.g. for subinertal, geostrophic flows), but some asymmetry is likely to be found. We will bring this point into the revised text.

**How is the value of R motivated?**

In the original manuscript we used a literature value for the linear drag coefficient. However, in response to comments from both reviewers, we will update the calculations in the manuscript using an effective linear friction coefficients $R = Cd*|u|$ instead, where $Cd$ is the model's quadratic drag coefficient and $|u|$ is an expected magnitude of current strength. By using the model's actual drag coefficient ($Cd= 0.0025$) and mean values for current strength $|u|$ (0.29 and 0.23 m/s for Mosken/Værøy and Røst, respectively), we obtain effective linear friction coefficients of values $R=8.6 \cdot 10^{-4}$ $m/$s and $R=5.7 \cdot 10^{-4}$ m/s for Mosken/Værøy and Røst, respectively. These are mean values calculated for 5 and 9 closed depth contours between 30 m and 50 m around Mosken/Værøy and between 30 m and 70 m around Røst, respectively.

**Closer to islands, D decreases. So, how appropriate is a choice D=50m?**

The slope, where we expect the main vorticity generation to occur, by either mechanism, is mainly located between 30 and 100 m, which can be seen from the bottom contours in Figure 1 below. These depths are therefore the most interesting to investigate regarding vorticity generation and rectification (compare with Fig. 12 of the manuscript). In Figure 1 below, we show a spatial calculation of $fD/R$. Here we used an effective linear friction coefficient $R = Cd*|u|$, where $|u|$ is the amplitude of current speed, independent of direction. We see that the $fD/R$ is mainly between 2 and 4 at the upper slope and more than 10 over the deeper parts of the slope. At very shallow depths near the islands, the bottom friction torque begins to dominate. However, here the slope is gentle, and the vorticity generation is expected to be weak. Therefore, we believe that 50 m is a reasonable choice for the scaling, and that vorticity generation by squeezing and stretching dominates in general around these slopes, and is most interesting to investigate in more detail. But, $fD/R$ is not much greater than 1 over slopes, hence, we do not believe the bottom friction torque is negligible—as we already state in the manuscript.

We did the same calculations around the same closed contours as mentioned above. Her we got on average value of fD/R ~ 6 around Værøy/Mosken (fD/R ranging from 5 to 8) and average value of fD/R ~ 12 around Røst (fD/R ranging from 7-19). We will consider adding this information in the revised manuscript.

[Figure]

*Figure 1 fD/R for the region around Mosken/Værøy and Røst. Here R = Cd\*|u|, and |u| is the amplitude in current speed, independent of direction. The contours show the bottom topography.*

**Line 519: Bottom intensification of tidally rectified flows and concomitant vertical circulation cells (Maas et al 1989) might possibly be of relevance for the transport and spreading of marginally sinking larvae and cod eggs.**
Thank you for the reference. We will include this in the discussion. Another related issue: the prominent tidal motion, particularly in interaction with topography, will also induce strong vertical mixing which may greatly reduce the stratification. So, the amount of bottom intensification is presumably influenced by small-scale mixing processes set up by the same currents that drive the mean flow. As mentioned in the manuscript, we intend to have a look at stratification effects in the 3D follow-up study and assessing bottom intensification would be a natural part of that.

**Figures A1 and A2 have interchanged captions**

Yes, thank you for pointing this out, we will fix this in the revised manuscript.

---

## Author Response (AR1)

**Response to reviewers:**

We thank both reviewers for encouraging yet constructive comments and critique. Below are point-by-point responses to their comments, along with references to line numbers for revised text in the updated version of the manuscript.

**Reviewer 1**

In this work the authors examine tracer dispersal due to tidal processes around the Lofoton-Vesterålen with the motivation of determining the importance of tides modulating the dispersal of cod eggs and larvae from spawning to nursery areas. They focus on two non-linear tidal processes, namely tidal pumping and tidal rectification. FVCOM is run with an unstructured grid 2D setup for tides only and subsequent tracer releases are analysed. The authors then analyse the relative importance of each process in driving the exchange of waters in and out of Vestfjorden through a number of different straits. The paper is well written, clear and nicely presented. My comments are generally related to the presentation of the results and how they could be made more easily to understand

**Introduction:**

**L.18–21: could do with a few additional references**

Good point. We have included two references on the acknowledged role of tides for vertical mixing.

Modified text in l. 19-21:
*While strong tidal currents are known to cause efficient vertical mixing of the ocean, important for bringing up nutrients in the water column (e.g. Blauw et al., 2012; Richardson et al., 2000), their contribution to net horizontal transport is often underestimated due to their oscillating nature.*

**L 50-65: It would be nice to have a diagram to illustrate this process. You could move Figure 9 to the intro and refer to it here as it greatly adds to the intuitive understanding of the process.**
**L 68-83: Again, it would be nice to have the diagram for this in the intro.**

This is an interesting suggestion. We have considered moving the two diagrams (Figs. 9 and 14) to the intro to help illustrating the two transport processes better. However, on further thought we decided against this since Fig. 9 contains details (definition of parameters), which, we feel does not belong in an introduction section.

So, we did not move the diagrams to the introduction. Instead, we landed on a compromise, adding references (in parentheses) to these two figures in the Introduction (l. 56 and 74).

**Methods:**

**L 118: Why do you use TPXO7.2? The latest version is 9 and is available as a higher resolution dataset. This is likely to affect your results. By how much do open boundary forcing values differ between the two datasets?**

The actual model run was conducted a few years back, but we got delayed in analyzing the data and writing this paper. This is why an older version of TPXO has been used. However, we believe that the validation against previous studies as well as observations is the most important check regarding model performance. We take the favorable comparisons shown in Figure 5 as indication that this model is adequate for investigating what are, first and foremost, qualitative aspects of the tidal dynamics and transport in the region.

Incidentally, Figure 5 has been updated after a bug was found in the validation script. The agreement is arguably better after the update except for the phase of the S2 tide (which, however, has a very small amplitude compared to other constituents).

Modified text in l. 141-144:
*Furthermore, the sea surface height and phase from the model fit reasonably well with observations from five stations provided by the Norwegian Mapping Authority, Hydrographic Service (2021), as shown in Fig.5. One notable exception is the phase of the S2 tide, but the amplitude of this is very small compared to the other constituents.*

**Model validation:**

**It would be nice to see some numerical values so that the reader can assess the performance of the model in absolute terms. Could you add a table with the amplitudes, phases, and amplitude and phase errors for each station for M2 and K1? You could also report mean errors for each constituent analysed.**

We have included two new tables (Tables 1 and 2) with the values from M2 and K1, corresponding to Fig. 5.  In these tables we have also included the 95% confidence interval for the sea surface height amplitude, but not for the current meter data since these statistics are not given in Moe et al 2002.

**Tidally driven transport:**

**L.177: Why do you show results only for 3 and 4.5 hours? It would be good to see the process through the tidal cycle at e.g. hourly intervals.**

We agree that a higher temporal resolution could be useful in these figures. However, there needs to be a compromise. Essentially, including many more time-steps will make each subfigure smaller and harder for the reader to interpret. The figures show that dipoles form and partly escape the return flow in these two straits, which is the main point. We landed on a compromise of adding two subplots, showing the situation additionally at 1.5 and 7.5 hours (after the first slack tide).

Modified text in caption, Figure 7:
*The two upper panels and the left middle panel are during northward flow (flood tide), and the right middle panel and the two lower panels are during southward flow (ebb tide).*

Modified text in l. 173-176:
*The various panels show the situation at various times after slack tide after ebb tide. So, the first three panels (1.5, 3 and 4.5 hours) show conditions during the flood tide while the last three (7.5, 9 and 10.5 hours) show conditions through the following ebb. Already at 1.5 hours after slack tide we*

*see that the northward-flowing tidal current has separated from the coast near the abrupt opening in the north.*

Modified text in l. 178:
*The vortices form a self-propagating dipole pair and grow in time, as can be seen at 3 hours and 4.5 hours.*

Modified text in l. 181:
*The ebb tide (7.5, 9 and 10.5 hours in the Fig. \ref{fig: Nappstraumen}) returns water to the northern opening as potential flow, following the shape of the coastline.*

Modified text in l. 190-193:
*A closer inspection shows that the dipoles form later in the tidal cycle compared to the generation at the northern exit in Nappstraumen (3 hours compared to 1.5 hours),*

**Rectified tidal transports:**

**L.335: I'm not sure this sentence fully makes sense? Which mechanism dominates and is responsible for the net flux or how does each process contribute?**

We agree with you here and have modified the text here in the revised manuscript. The main message that we try to get across is that tidal pumping is the dominating mechanism causing net fluxes out of Vestfjorden in general. But south of Røst the net flux is mainly due to tracer advection by the rectified tidal currents (not tidal pumping).

Modified text in l. 330-333:
*So even though much of the net tracer transport south of Lofotodden is due the tidal pumping mechanism investigated above, there is also a contribution driven by anticyclonic mean flows around the islands here. This mechanism appears to be particularly important south of Røst where, it should be noted, there can be no formation of self-propagating dipoles*

**L343: How will your choice of linear bottom friction impact your results? Is this the same bottom friction used in the tide model? It may be an issue if they are not consistent?**

The numerical model uses quadratic bottom friction whereas in our discussion of theory we have used linear bottom friction in order to arrive at closed-form analytical expressions (eqns. 12 and 13).

As for comparison between simplified theory and numerical results (with regards to the response to the spring-neap cycle), we primarily seek an order-of-magnitude agreement since the theoretical expression is obviously simplified. The reviewer's comments nevertheless led us to improve on our previous estimate for an effective linear drag coefficient. Essentially, we set $R = C_d*|u|$, where $C_d$ is the model's quadratic drag coefficient and $|u|$ is an expected magnitude of current strength.

As outlined in the text, we investigated 5 and 9 closed depth contours around Mosken-Værøy and around Røst, respectively. Updating these calculations, now using the model's actual drag

coefficient ($C_d$= 0.0025) and mean values for current strength |$u$| (0.29 and 0.23 m/s for Mosken/Værøy and Røst, respectively), led to an updated Figure 13.

Here, we found effective linear friction coefficients to be $R = 8.6 \cdot 10^{-4}\, m/s$ and $R = 5.7 \cdot 10^{-4}\, m/s$, for Mosken/Værøy and Røst, respectively. These are mean value for all the closed depth contours around each of the island groups. Using these updated values in our expression for phase lag,

$$\phi = tan^{-1}\left(\frac{\omega_{sn}H}{R}\right),$$

we obtain an updated mean phase-lag of the response to the spring-neap cycle of 0.6 days (0.24 rad/s) and 0.9 days (0.39rad/s), for Mosken-Værøy and for Røst, respectively.

Modified text in l. 367-374:

*The primary slow time scale variation in forcing for our problem is the spring-neap cycle. So $\omega_{sn}$= 2π/14.75 rad days$^{-1}$. To test the theory with respect to this variation we additionally need to specify a depth level H and a linear friction coefficient R. Our FVCOM model uses quadratic bottom drag, but an equivalent linear drag coefficient can be found from R=Cd|u|, where Cd= 0.0025 (the value used in the model) and |u| is a typical current strength. We diagnosed values of 0.29 and 0.23 m s$^{-1}$ for the current strength around Mosken–Værøy and Røst, respectively, and used these to calculate equivalent linear friction coefficients. Then taking a typical depth where the slope is steep, H = 50m, we calculate theoretical spin-up times of approximately 21 and 14 hours which corresponds to a phase lag φ of about 0.39 and 0.24 radians, for Mosken–Værøy and Røst, respectively.*

**L375: How does the choice of filter impact the curves? It would be nice to see 1 tidal cycle values as well.**

The purpose of the low-pass filter is to make the figure cleaner by removing the dominating diurnal and semi-diurnal variations. But we found that it was possible to add these with thinner lines—and updated Figure 13 accordingly (reproduced below).

[Figure]

*L440: I'm not sure I understand how you define the topographic length scale*
*L445: Where do you define $L_B$ and what is it? How does it differ from L?*

We see that we have forgotten to define the topographic length scale in the manuscript and thank the referee for pointing this out.

$L_B$ is the topographic length scale, defined as follows:
$$L_B = \frac{H_0}{\alpha}, L_B = \frac{H_0}{\Delta H}$$
where $H_0$ is a mean depth and $\alpha = \Delta H$ is the slope of the bottom topography. $L_B$ is thus used to characterize bathymetric feature, which will cause water to move vertically over a ridge or a bump etc. at the sea floor bottom. $L$ on the other hand is the horizontal distance the water column has moved within some time dictated by a given dynamical process. In our case here, $L = L_T$, where $L_T$ is half a tidal cycle.

*We have now included the definition of topographic length scale in the manuscript in lines 446-47: A topographic length scale Lb = H0/ΔH can then be defined as that which gives a depth excursion equal to the initial depth, or H0 = αLB.*

**Summary and conclusion:**

**L.492: 'But not all straits are created equal' – reword to something more specific.**

This was a play on words inspired by wording in the American Declaration of Independence. We have reworded this general transitional sentence, but we leave specifics to the sentences that complete that paragraph.

Modified text in l. 499-500:
*But geometry and flow conditions around each strait are different, and the tracer transport due to tidal pumping varies greatly.*

**General: Given that your introduction talks about cod quite a lot, the reader feels a little let down in the conclusions. How important are these tidal processes for the dispersal of the Arctic cod?**

This is a good point. Since we only investigate purely tidally driven flows in this manuscript, we cannot say too much about the contribution relative to other processes that impact the drift of cod eggs out of Vestfjorden (winds, background currents, freshwater run off etc.). For such an assessment we point towards our more complete 3D follow-up study at the very end of the Conclusions section. But, to not promise more than we can keep in this manuscript, we have now downplayed the focus on cod-eggs in the introduction.

Modified (shortened) text in l. 24-29:
*In this study, we will investigate nonlinear tidal dynamics around Lofoten-Vesterålen in Northern Norway (Fig. 1), a major spawning ground for the Northeast Arctic cod (Hjermann et al., 2007). Spawning of this species takes place all along the middle and northern Norwegian coast, but as much as 40 percent of the cod spawns in Vestfjorden southeast of the Lofoten-Vesterålen archipelago (Ellertsen et al., 1981; Sundby and Bratland, 1987). Therefore, a good understanding of ocean dynamics controlling the drift and spreading patterns of biogeochemical material, and cod eggs and larvae in particular, is important for identifying particularly vulnerable regions and factors controlling the recruitment of the Northeast Arctic cod.*

**How do the observed processes compare to other regions in the world and different observations? It would be nice to see this work be put into context of other previous work in this section.**

This is a useful suggestion. We have now put the work a bit more into context of previous work in the revised manuscript, mainly in the Conclusions section.

Modified text related to tidal pumping in l. 504-511:
*Tidal pumping, particularly in relation to tidal flushing of estuaries and near-shore regions, have been widely studied elsewhere. Certainly, the formation of dipole vortices is observed many places where prominent tidal currents exit narrow straits, for example in Aransas Pass (USA), Messina Strait (Italy) and the Great Barrier Reef (Australia) (Whilden et al., 2014; Cuccoet al., 2016; Delandmeter et al., 2017). Cucco et al. (2016) show that the strong tidal currents and subsequent pumping is important for water exchange and for modifying the thermohaline properties in two large sub-basins of the Western Mediterranean Sea. We thus consider it possible that tidal pumping in Lofoten and Vesterålen not only contributes to transport of dynamically passive particles such as cod eggs but is also important for the transport of freshwater out of the large Vestfjorden embayment, thereby modifying the thermohaline properties here.*

Modified text related to tidal rectification in l. 522-527:
*We find that the potential for tidal rectification can be evaluated through the relation $L_T/L_B$, where values near one indicate prominent rectification (Loder, 1980). The residual currents around the island groups in southern Lofoten thus appear to be governed by dynamics similar to what is observed around Georges bank (Loder, 1980; Limeburner and Beardsley, 1996; Chenet al., 2001). There, the residual currents encircling the bank in a clockwise fashion are of similar strength as the flow we model around Røst and Mosken/Værøy (0.2–0.3 m/s). Using the values provided in table 1 in Loder (1980), we find that on the northwestern and northern side of Georges Bank $L_T/L_B \sim 1$, equivalent to what we find for the islands of southern Lofoten.*

**Figures:**

Here we will only shortly comment on the direct questions. Suggestions of technical improvements and rephrasing figure captions have all been implemented in the revised manuscript.

**Figure 2: It would be nice to have more information with this figure: e.g. (a) how do the two figures differ? (I guess it's the phase of the tide); (b) the velocity front is identified – why is this relevant and what does it mean for this study?**

The purpose of this figure is to show that formation dipoles and tidal jet are actually observed in Moskstraumen, and not merely a result of numerical modelling. But we have added some information to guide the reader.

The caption now reads:
*Satellite images from Copernicus Sentinel-II missions, tracing out surface currents in Moskstraumen and Nordlandsflaget. The left panel shows the current structure during westward flow (during flood tide); west of Lofotodden a velocity front is evident which is likely related to dipole formation. The right panel shows the current structure during eastward flow; here, a dipole east of the strait with a trailing jet is evident. The Sentinel-II missions satellites carry a multi-spectral instrument with 13 spectral channels in the short-wave infrared and visible/near infrared spectral range,*

*whereas this image is collected from band B4 (664.6 nm). The satellite imagery was assessed and processes using data from the Norwegian National Ground Segment for Sentinel data (Halsne et al., 2019, pers. comm. Trygve Halsne).*

**Figure 4: how about plotting observed values on top as shaded circles on top of the model results? Also, plot against latitude and longitude rather than distance.**

The suggestion of plotting observed values on top of the model results is a good idea. However, the observations are typically located near land, e.g. inside straits, where gradients are large. We thus decided against adding observations in this figure, both because adding shaded circles would make the figure messy and potentially hide the overall structure of the tides, and because model-observation comparisons are presented in Fig. 5 and (new) Tables 1 and 2.

We have plotted against latitude and longitude but did not add observations to the figure.

**Figure 6, l.5: do = does**

Fixed.

**Figure 7: Could you take your panels a bit further south? At 10.5 hours you are missing part of the incoming waters**

We eventually decided against this since the water is flowing out of the southern opening at 10.5 hours. The textual focus here is on the dipole on the northern side.

**Figure 9: Panel b: Why is $A_e$ in the sink region? Should it not be in the jet? It would be nice to illustrate all of the length scales on this plot (as far as possible)**

$A_e$ is in the jet (the black two-headed arrow). But we see that the location of the label was poorly placed.

We have now moved the label further south and included an arrow showing the sink radius.

**Figure 10: Not all marker labels are visible. Also, you could increase the intuitiveness of the plot by adding labels such as an arrow with 'more asymmetric' along the axis y axis and doing the same for the x-axis.**

We have adjusted labels so that they are now all visible. We have also added 'help' arrows, as suggested.

**Figure 11: Mention what each non-dimensional parameter is in words either on the axis or in the legend.**

Done. The new caption is now:
*The tracer transport efficiency $T^*_p$ plotted against non-dimensional parameters (a) Axo, representing flow asymmetry, (b) $L^*$ representing strait length and (c) AxoL*, combining the two non-dimensional parameters. Two estimates of Axo are shown for Gimsøystraumen (5).*

**Figure 13: What do you mean by 'sets of closed contours'? Could you show these on a map?**

*The "sets of closed contours' should have been phrased as "sets of 5-9 closed depth contours encircling the island groups". We have now included a map similar to the one in Figure 17 for the readers to see the locations of these contours.*

**Figure 14: It would be nice to add the positive/negative vorticity gains in the diagram – it would make it even more intuitive. You could also explicity include the words squeezing and stretching in the diagram.**

*The figure has now been modified according to these suggestions.*

Minor comments:

**Generally, watch out for the use of commas. E.g. Here, … In this study,…**
**l.16: rises = raises**
**l.54: is = are**
**59: what = which**
**102: no comma needed**
**425: 'is down the' = is down to the**
**520: appears = appear**

Thanks. We have edited the manuscript according to all of the comments, and a check of comma use has been done.

**Reviewer 2**

This paper discusses two physical processes, tidal pumping and tidal rectification, that may be relevant to the dispersion and transport of cod eggs and larvae in the Lofoten - Vesteralen area, Northern Norway.

The geographical and ecological settings, as well as the physical processes and numerical techniques used to address the questions involved in larvae and cod egg transport and spreading are well described. The paper comes to a clear conclusion regarding the relevance of tidal pumping, and provides welcome suggestions for further investigation of the process of tidal rectification.

As a consequence, it is recommended that the paper is published provided it addresses the few minor issues discussed next.

**Line 404: The along-isobath velocity scale U may be much stronger than the cross-isobath current used in the previous scaling (line 403).**

Good point. The amount of asymmetry between the two velocity components (along and across topography) will depend on both time and length scales of the problem. For high-frequency tidal oscillations the cross-isobath component need not be severely constrained (as it will be e.g. for subinertal, geostrophic flows), but some asymmetry is likely to be found. We actually looked into the model and found the ratio between along- and across-slope velocity components around the island groups to be 1.2–1.4.

We have included a small discussion on this in l. 409-415:

*Comparing the magnitude of the two terms, using typical values for Mosken-Værøy and Røst as above (D ~50 m, R=6·10⁻⁴ and 9·10⁻⁴ m s⁻¹ and f ~10⁻⁴ s⁻¹), gives $fD/R$ ~6 and 12 for Mosken-Værøy and Røst, respectively. Here, we have picked a depth value which corresponds to the steeper parts of the slope (where vorticity generation by either mechanism can be assumed to be most relevant) and assumed that the along-slope and cross-slope velocity components are of similar magnitude. One might intuitively expect the along-slope component to be larger than the across-slope component, but perhaps primarily for longer-timescale (subinertial) motions. Diagnosing the model fields around Mosken-Værøy and Røst showed that the ratio between the two is only about 1.2–1.4 for the tidal motions considered here (calculated for the depth contours in Fig. 13). This suggests that vorticity production by flow up and down topography is quite a bit larger than production by bottom friction torque.*

**How is the value of R motivated?**

In the original manuscript we used a literature value for the linear drag coefficient. However, in response to comments from both reviewers, we have updated the calculations in the manuscript using an effective linear friction coefficients $R = C_d*|u|$ instead, where $C_d$ is the model's quadratic drag coefficient and $|u|$ is an expected magnitude of current strength. By using the model's actual drag coefficient ($C_d= 0.0025$) and mean values for current strength $|u|$ (0.29 and 0.23 m/s for Mosken/Værøy and Røst, respectively), we obtain effective linear friction coefficients of values $R$=8.6·10⁻⁴ m/s and $R$=5.7·10⁻⁴ m/s for Mosken/Værøy and Røst, respectively. These are mean values calculated for 5 and 9 closed depth contours between 30 m and 50 m around Mosken/Værøy and between 30 m and 70 m around Røst, respectively.

Modified text in l. 367-374:

*The primary slow time scale variation in forcing for our problem is the spring-neap cycle. So $\omega sn$= $2\pi/14.75$ rad days⁻¹. To test the theory with respect to this variation we additionally need to specify a depth level H and a linear friction coefficient R. Our FVCOM model uses quadratic bottom drag, but an equivalent linear drag coefficient can be found from $R=C_d|u|$, where $C_d$= 0.0025 (the value used in the model) and $|u|$ is a typical current strength. We diagnosed values of 0.29 and 0.23 m s⁻¹ for the current strength around Mosken–Værøy and Røst, respectively, and used these to calculate equivalent linear friction coefficients. Then taking a typical depth where the slope is steep, H = 50m, we calculate theoretical spin-up times of approximately 21 and 14 hours which corresponds to a phase lag $\varphi$ of about 0.39 and 0.24 radians, for Mosken–Værøy and Røst, respectively.*

**Closer to islands, D decreases. So, how appropriate is a choice D=50m?**

In Figure 1 below, we show a spatial calculation of $fD/R$. Here we used an effective linear friction coefficient $R = C_d*|u|$, where $|u|$ is the amplitude of current speed, independent of direction (as outlined in our response to the previous comment). At very shallow depths, less than 20-30 m, the bottom friction torque begins to dominate. However, here the slope (inclination) is generally gentle, and vorticity generation is expected to be weak. The steeper parts of the slope, where we expect the main vorticity generation to occur, by either mechanism, is mainly located between 30 and 100 m (seen from the bottom contours). These depths are therefore the most interesting to investigate regarding vorticity generation and rectification (compare with Fig. 12 of the manuscript). Therefore, we believe that 50 m is a reasonable choice for the scaling, and that vorticity generation by squeezing and stretching dominates in general around these slopes. But,

*fD/R* is not much greater than one over the slopes; hence, we do not believe the bottom friction torque is negligible—as we already state in the manuscript.

[Figure]

*Figure 1 Estimates of fD/R for the region around Mosken/Værøy and Røst. Contours show the bottom topography.*

We have included a small comment on this point in l. 409-415 (same text as for comment on anisotropic velocities above).

**Line 519: Bottom intensification of tidally rectified flows and concomitant vertical circulation cells (Maas et al 1989) might possibly be of relevance for the transport and spreading of marginally sinking larvae and cod eggs.**

Thank you for the reference. Another related issue: the prominent tidal motion, particularly in interaction with topography, will also induce strong vertical mixing which may greatly reduce the stratification. So, the amount of bottom intensification is presumably influenced by small-scale mixing processes set up by the same currents that drive the mean flow. As mentioned in the manuscript, we intend to have a look at stratification effects in the 3D follow-up study and assessing bottom intensification would be a natural part of that.

The Maas and Zimmermann (1989) reference has been included in a general listing of possible 3D effects, in l. 544-549:
*Our simulations were also limited by their 2D nature. A 2D configuration was chosen to help isolate nonlinear lateral tidal dynamics, but the model was thus unable to account for baroclinic effects. Such effects include the generation of hydraulic jumps and vertical mixing around strait openings (Lynge et al., 2010), the establishment of density fronts around the rectification cells (Ou, 2000) and also bottom intensification of such rectified flows, with concomitant vertical circulation cells around banks and islands (Maas and Zimmerman, 1989; White et al., 2005, 2007). So, in reality, baroclinic flow dynamics will also impact tracer transport, both vertically and laterally.*

**Figures A1 and A2 have interchanged captions**

Yes, thank you for pointing this out. We have now fixed this.

**Line 53: waters => water**
**Caption Fig. 2: instruments => instrument**
**Line 127: drop 'the'**
**Line 443: chances => changes**
**Line 462: dynamically => dynamical**
**Line 519: Bottom intensification of tidally-**

We have fixed all these mistakes in the revised manuscript.

---

## Author Response (AR2)

Thanks very much to the Editor for a last set of comments. Below are responses to each of these. We also found a few mistakes ourselves that we have corrected, which you will find at the end of this document.

Line 54: Delete "clearly" *- Fixed*

Paragraph from line 47: I agree with the reviewer that the diagrams would help a lot at this early stage, especially as Fig 1 doesn't include the tidal pumping mechanism. Please move fig 9 and fig 14 to this paragraph (they could be combined as two panels of one figure if you prefer). I understand your argument about the notation but it is easier to refer back to a diagram on the first or second page having already grasped the physical picture than forwards several pages.

*- Done. The sketches are now in the Introduction section, in two separate figures to avoid too long figure caption.*

Line 122 It might not affect the results, but what phase was the tide at the point of tracer release?

*- We have now included this information in the manuscript (l. 124-125).*

Fig 4 top right the label K1 has gone wrong. *- Fixed*

Equation 4 is incorrect, a font problem I think. Please check. *- Fixed*

The fit in Fig 10 is a bit unconvincing, especially on a log-log axis. But I don't think it necessarily damages your later argument.

*- We were a bit unsure what we should implement from this comment. In the end, we kept the log-log presentation—to retain equal 'resolution' for values larger and smaller than 1—but downplayed the interpreted relationship in the corresponding text (l. 268-269).*

Fig 13 needs a legend. Also "encricling"-> "encircling" in the caption. *- Fixed*

Line 420: Tidy. "Even though the latter term is not necessarily negligible, we will omit it in the following for simplicity..." *- Fixed*

Fig 15: If you haven't already, please check this colour scheme: https://www.color-blindness.com/coblis-color-blindness-simulator/

*- Thank you for pointing this out. We have now changed the colormap in this figure.*

Fig 17 needs a legend. *- Fixed*

I hate to say it but "data available on request" is not enough, and not much help to a researcher in ten years time if you have a new job and don't answer your emails! Please consider uploading model results to a public archive server. If uploading the whole model run is impossible, you should at least make public the data necessary to reproduce the figures. See https://www.ocean-science.net/submission.html#assets

*- The data set has now been uploaded and made public through a Norwegian national HPC storage service (NIRD): (https://archive.sigma2.no/pages/public/datasetDetail.jsf?id=10.11582/2021.00095 )*

*The information has been entered under "Data availability".*

Data used for model boundary conditions must be included in Acknowledgements, using the data statement as provided with the data, as should the satellite images (people who fund satellites like to run searches to see their money is being well spent). And the bathymetry. Oh, and the tide stations in figure 5 - where is their data hosted please? *- Fixed*

*Additional corrections (spotted by ourselves and Dmitry Aleynik):*

*Fig. 9 and Fig.10 (prev. Fig 7 and 8) : description of streamlines is included in the caption.*

*Fig. 15: In caption, black contour lines are corrected to white contour lines for bottom topography.*

*Eq 10-11: 1/H is moved into the integral for consistency with Eq. 9 and 12.*

*L.231 : We included a sentence on neglecting the Coriolis acceleration in eqn. 3 (truncated momentum).*

*L. 277 : eastern (ebb) opening of Nordlandsflaget*

*L. 345: Changed use of Stokes theorem to use of Green's and Gauss' theorems.*

*L. 351-352: Added a sentence explaining why fu is neglected in Eq.11*

*Eq. 14: Took out f from D/Dt since D/Dt(f) = 0 (makes it less messy).*

*Eq. 18: the left side should be divided by h^2 not just h (fixed)*